

**A new step-wise Carbon Cycle Data Assimilation System**
**using multiple data streams to constrain the simulated land**
**surface carbon cycle**
**P. Peylin[1], C. Bacour[2], N. MacBean[1], S. Leonard[1], P. J. Rayner[1,3], S. Kuppel[1], E.**
**N. Koffi[1], A. Kane[1], F. Maignan[1], F. Chevallier[1], P. Ciais[1], P. Prunet[2]**
[1]{Laboratoire des Sciences du Climat et de l'Environnement, UMR 8212 CEA-CNRS-
UVSQ, 91191 Gif-sur-Yvette cedex, France}
[2]{Noveltis, Parc Technologique du Canal, 2 avenue de l'Europe, 31520 Ramonville-Saint-
Agne, France}
[3]{University of Melbourne, 3010, Vic, Melbourne, Australia}
Correspondence to: P. Peylin (philippe.peylin@lsce.ipsl.fr)





**Abstract**
Large uncertainties in Land surface models (LSMs) simulations still arise from inaccurate
forcing, incorrect model parameter values and incomplete representation of biogeochemical
processes. The recent increase in the number and type of carbon cycle related observations,
including both in situ and remote sensing measurements, has opened a new road to optimize
model parameters via robust statistical model-data integration techniques, in order to reduce
the simulated carbon fluxes and stocks uncertainties. In this study we present a Carbon Cycle
Data Assimilation System (CCDAS) that assimilates three major data streams, namely
MODIS-NDVI observations of vegetation activity, net ecosystem exchange (NEE) and latent
heat (LE) flux measurements at more than 70 sites (FLUXNET), and atmospheric $CO_2$
concentrations at 53 surface stations, in order to optimize the main parameters of the
ORCHIDEE LSM (around 180 parameters in total). The system relies on a step-wise
approach that assimilates each data stream in turn, propagating the information gained on the
parameters from one step to the next.
Overall, the ORCHIDEE model is able to achieve a consistent fit to all three data streams,
which suggests that current LSMs have reached the level of development to assimilate these
observations. The assimilation of MODIS-NDVI (step 1) reduced the growing season length
in ORCHIDEE for temperate and boreal ecosystems, thus decreasing the global mean annual
gross primary production (GPP). Using FLUXNET data (step 2) led to large improvements in
the seasonal cycle of the NEE and LE fluxes for all ecosystems (i.e., increased amplitude for
temperate ecosystems). The assimilation of atmospheric $CO_2$, using the atmospheric transport
model LMDz (step 3), provides an overall constraint (i.e., constraint on large scale net $CO_2$
fluxes), resulting in an improvement of the fit to the observed atmospheric $CO_2$ growth rate.
Thus the optimized model predicts a land C sink of around 2.2 $PgC.yr^{-1}$ (for the 2000-2009
period), which is more compatible with current estimates from the Global Carbon Project
(GCP) than the prior value. The consistency of the step-wise approach is evaluated with back-
compatibility checks. The final optimized model (after step 3) does not significantly degrade
the fit to MODIS-NDVI and FLUXNET data that were assimilated in the first two steps,
suggesting that a stepwise approach can be used instead of the more "challenging"
implementation of a simultaneous optimization in which all data streams are assimilated
together. Most parameters, including the scalar of the initial soil carbon pool size, changed



during the optimization with a large error reduction. This work opens new perspectives for
better predictions of the land carbon budgets.

## 1 Introduction

Atmospheric $CO_2$ concentrations have increased at an unprecedented rate over the last few
decades, predominantly due to anthropogenic fossil fuel and cement emissions, as well as
land use and land cover change (LULCC). The oceans and the terrestrial biosphere have
absorbed $CO_2$, removing on average 50% of anthropogenic emissions from the atmosphere.
However, knowledge about the exact location of sources and sinks of carbon (C) and the
driving mechanisms is still lacking. Land surface models (LSMs) can be used to improve our
understanding of the spatio-temporal patterns of sources and sinks, as well as for attributing
changes due to $CO_2$, climate variability and other environmental drivers. However, the spread
in the model predictions of terrestrial net C exchange currently has the same order of
magnitude as the uncertainty of the terrestrial C budget estimated as the residual of the other
components (Le Quéré et al., 2015). In addition to uncertainties in the mean global annual
terrestrial C budget and its trend over time (Sitch et al., 2015), there remains strong
discrepancies between LSMs in their predictions of regional budgets (Canadell, 2013) at
seasonal and inter-annual timescales and in their sensitivity to climate and atmospheric $CO_2$
forcing (Piao et al., 2013).
Uncertainties in model simulations arise from inaccurate forcing, incorrect model parameter
values and/or an inadequate or incomplete representation of biogeochemical processes in the
model (for example the impact of nutrient limitation on C fluxes, or C release related to
permafrost thawing). Arguably the best way to improve model predictions is to confront
simulations with multiple sources of data within an appropriate and rigorous framework
(Prentice et al., 2015). In the last two decades significant efforts by the site and satellite
observation communities have resulted in a large increase in the number and type of C cycle-
related observations. These data contain some information at various spatial and temporal
scales and should be combined together to robustly address different aspects of the models.
One way in which these data can be used to better quantify and reduce model uncertainty is
by optimizing or calibrating the model parameters via robust statistical model-data fusion (or
data assimilation – DA) techniques. In particular a Bayesian inference framework allows us to



update our prior knowledge of the parameters based on new information contained in the
observations.
There is a long history of using DA techniques for parameter optimization, particularly in
Geophysics (Tarantola, 1987), but the initial studies in the field of global terrestrial C cycle
data assimilation started with the initial study of Fung et al. (1987) and a pioneering work by
Knorr and Heimann (1995) who used atmospheric $CO_2$ concentration to constrain the Simple
Diagnostic Biosphere Model (SDBM). This effort was continued by the original Carbon
Cycle Data Assimilation System (CCDAS) described in Rayner et al. (2005) and Kaminski et
al. (2012) which used both atmospheric $CO_2$ and satellite-derived Fraction of Absorbed
Photosynthetic Radiation (FAPAR) data to optimize vegetation productivity by adjusting the
C cycle-related parameters of the Biosphere Energy-Transfer Hydrology (BETHY) model
(see a review in Kaminski et al., 2013). Meanwhile substantial efforts have been put into the
use of local eddy covariance flux tower measurements of net exchange of $CO_2$ and latent and
sensible heat fluxes to optimize photosynthesis, respiration and energy-related parameters of
terrestrial ecosystem models, both at individual sites (e.g. Wang et al., 2001, 2007; Williams
et al., 2005; Braswell et al., 2005; Knorr and Kattge, 2005; Moore et al., 2008; Ricciuto et al.,
2008), and more recently using multiple sites together (hereafter multiple sites) from the
global FLUXNET network (e.g. Groenendijk et al., 2011; Kuppel et al., 2012, 2014; Alton,
2013; Xiao et al., 2014). Increasingly the focus in carbon cycle data assimilation is moving
towards using multiple different data streams as independent constraints, with the aim of
bringing more information at different spatial and temporal scales and constraining several
processes at once in order to reduce the likelihood of model equifinality (where multiple sets
of parameters achieve the same reduction in model-data misfit). Recent examples include the
combination of in-situ eddy covariance flux observations and ground-based information on
vegetation structure and C stocks (Richardson et al., 2010; Ricciuto et al., 2011; Keenan et al.,
2012, 2013; Thum et al., 2015), or in-situ flux data and satellite FAPAR (Kato et al., 2013;
Zobitz et al., 2014; Bacour et al., 2015). This is a non-trivial task however, especially when
optimizing a complex LSM (see MacBean et al, submitted), which has many parameters
acting from local to global scales.
When assimilating multiple different data streams we have two options: i) to optimize the
model with each data stream in turn, and to propagate the information gained on the
parameter values from one step to the next (hereafter referred to as "stepwise" assimilation),





or ii) to include all data streams together in the same optimization (hereafter referred to as
"simultaneous" assimilation). Kaminski et al. (2012) suggested that it is essential to perform a
consistent, simultaneous assimilation that includes all data streams in the same optimization.
It is important to note that this is an implementation question. Tarantola (2005) recasts the
fundamentals of the approach as the conjunction or multiplication of probability densities.
This multiplication is associative so it makes no difference whether it is performed in one step
or several. In complex problems such as these, one cannot carry or even describe the full
structure of the relevant probability densities so which approach will work best in each case is
unclear. In particular, technical difficulties associated with the different number of
observations for each data stream and the characterization of error correlations between them,
in addition to computational constraints to run global LSMs, might result in the preference for
a step-wise assimilation framework. Additionally, it may be more straightforward, to expose a
restricted set of parameters to each observation type in a stepwise approach to ensure that
each data stream constrains only the most relevant parts of the model. This reduces biases
from other poorly-represented processes caused by inadequate model structure. For these
reasons we follow the stepwise approach in this paper.
We present the first global-scale CCDAS that assimilates three of the main global data
streams that have been used to date to understand the terrestrial carbon cycle – atmospheric
$CO_2$ concentration, satellite-derived information of vegetation greenness (from the MODIS
instrument) and multisite eddy covariance net $CO_2$ and latent heat flux measurements (from
FLUXNET) – to optimize the parameters of the Organizing Carbon and Hydrology in
Dynamics Ecosystems (ORCHIDEE) process-based LSM (Krinner et al., 2005). The main
questions that we aim to answer in this paper are as follows:
i) How and to which extend the optimization of the ORCHIDEE model allows to fit the three
data streams that are considered?
ii) Does the step-wise optimization result in a degradation of the fit to other data streams used
in the previous steps?
iii) What are the main changes in the optimized parameters when using sequentially these
three data streams in a global CCDAS and which processes are constrained?
iv) What are the improvements for the land C cycle in terms of net/gross fluxes and stocks as
a result of multi-data stream optimization? What preliminary perspectives can we draw that





may help us in improving model predictions of trends, variability and the location of
terrestrial C sources and sinks?
Following these objectives, the paper first describes the new ORCHIDEE-CCDAS including
the concept, the observations, the models and the optimization approach. We then present the
results, including the fit to the data, consistency checks (question i) above) as well as mean
global and regional C cycle budget for the period 2000-2009. The last section discusses issues
and perspectives associated with these results.

## 9    2   Methods

### 10    2.1   ORCHIDEE-CCDAS concept

We have designed a CCDAS around the ORCHIDEE land surface model (ORCHIDEE-
CCDAS, later also referred to as ORCHIDAS for simplicity) that combines a state-of-the-art
description of the driving biogeochemical processes within the model with multiple
observational constraints in a robust statistical framework, in order to improve the simulation
of land carbon fluxes and stocks. The system allows us to retrieve the best estimate, given the
observations and prior information, of selected parameters (see §2.3.3) as well as to evaluate
their uncertainty. It relies on a stepwise assimilation of a comprehensive set of three C cycle-
related observations that are representative of small (100 m) to large (continental) scales (see
§2.2):

20       •   Step 1: Satellite measurements of vegetation activity using the Normalized Difference

21            Vegetation Index (NDVI) from the MODIS instrument over the 2000-2008 period for

22            a randomly selected set of sites for boreal and temperate deciduous vegetation types;

23       •   Step 2: In-situ eddy-covariance net $CO_2$ and water (latent heat) flux measurements

24            from the FLUXNET database for a large set of sites, spanning 7 different vegetation

25            types;

26       •   Step 3: In-situ monthly atmospheric surface $CO_2$ concentration measurements from

27            the GLOBALVIEW-CO2 database over three years (2002-2004).

28    The system relies on two models:

29       •   The ORCHIDEE global LSM, whose main C cycle parameters are optimized (see

30            §2.3)



•   The atmospheric transport model, LMDz (see §2.3), to relate the surface carbon fluxes

2        to atmospheric $CO_2$ concentrations.

The framework combines the different observational data streams within ORCHIDAS in
order to optimize selected model parameters using a variational data assimilation system,
described in section 2.4. Figure 1 illustrates the structure of the CCDAS and the different
components that are involved. Such a framework distinguishes i) the assimilated observations,
ii) an ensemble of forcing and input data streams, iii) the models and optimization framework,
as well as iv) an evaluation step, where independent datasets are compared to the optimized
model stocks and fluxes. As explained in the introduction, a major feature of the current
system is the stepwise approach, in which all data streams are assimilated sequentially (i.e.
one after the other). The information retrieved at a given step (retrieved optimal parameter
values and associated uncertainty) is propagated to the next step (see Fig. 2 and §2.4). Note
that for simplicity we did not propagated the error correlations in this first implementation of
the system.
At each step, the parameter optimization relies on a Bayesian framework that explicitly
minimizes the difference between the simulated and observed quantities in addition to
minimizing the difference between the optimized model parameters and "a priori" values (see
§2.4.2). The dependence of the simulated quantities on the optimized variables is non-linear,
which thus necessitates the use of an iterative algorithm. Note that all components of the
surface C budget need also to be included in the ORCHIDAS, particularly when using
atmospheric $CO_2$ measurements which requires the atmospheric transport model to be
prescribed with fossil fuel emissions, $CO_2$ fluxes associated with biomass burning and ocean
$CO_2$ fluxes (see §2.5) in addition to net ecosystem exchange (NEE) from ORCHIDEE.
**2.2   Assimilated observations**
2.2.1   MODIS-NDVI
MODIS collection 5 obtained from surface reflectance data (from 2000-2008) in the red (R)
and near-infrared (NIR) bands at 5 km resolution (CMG) are used to optimize the phenology-
related parameters of ORCHIDEE in the first step. The R and NIR data were processed to
correct for directional effects following Vermote et al. (2009) and then used to calculate the
NDVI, which is assumed to be linearly related to the model FAPAR. The NDVI are then i)
aggregated to the 0.72° spatial resolution of the ERA-Interim meteorological fields that are



used to force ORCHIDEE, ii) interpolated to a daily time series and iii) checked for quality
(see MacBean et al., 2015 for details). If there is a gap in the observations of more than 15
days, no interpolation is done (i.e., no data during the gap are assimilated). Figure 3 displays
the location of the sites that were selected (see §2.4.1).

### 2.2.2 Eddy covariance flux data

Eddy covariance flux measurements of net surface $CO_2$ flux – hereafter referred to as net
ecosystem exchange (NEE) and latent heat flux (LE) from 78 observation sites of a network
of regional networks (FLUXNET; see Fig. 3) are used to constrain ecosystem physiology and
fast C-related processes at daily to seasonal timescales in ORCHIDEE in the second step. We
use quality-checked and gap-filled data from a global synthesis called the La Thuile dataset
(Papale, 2006). In order to avoid dealing with the large error correlations in the half-hourly
data (see Lasslop et al., 2008), daily mean values of NEE and LE are used in the ORCHIDAS.
Days with less than 80% of the half-hourly data are left out of the assimilation. The selection
of the sites and the data processing (gap-filling, correction for energy balance closure) are
detailed in Kuppel et al. (2014).

### 2.2.3 Atmospheric $CO_2$ concentrations

Atmospheric $CO_2$ concentration measurements were taken from an ensemble of selected
surface stations around the world (Fig. 3). The spatial concentration gradients relate to the
integral of the fluxes over large areas and thus allow the optimization of large-scale global
patterns of carbon fluxes. These data were taken from the NOAA Earth System Laboratory
(ESRL) GLOBALVIEW-CO2 collaborative product (GLOBALVIEW-CO2, 2013) and
averaged to monthly means. We assimilated the monthly values for 53 sites for the 2002-2004
period inclusive in the last step of the assimilation system. Such restricted period (3 years
only) was chosen for practical reasons (computing resources) while constructing the
ORCHIDAS system. The station locations, indicated in Fig. 3, favor the background
conditions i.e. the surrounding air masses are only weakly influenced by local continental
sources, such as power plants. The choice of monthly mean is related to the use of pre-
calculated transport fields with LMDZ (see §2.3.2).



**2.3    Models and optimized parameters**
2.3.1  ORCHIDEE land surface model
In this study we use the ORCHIDEE process-oriented land surface model (Krinner et al.,
2005), which computes water, carbon and energy balances at the land surface on a half hourly
time step, using a mechanistic description of the physical and biogeochemical processes (see,
http://labex.ipsl.fr/orchidee/). The model describes the exchange of carbon and water at the
leaf level, the allocation of carbon within plant compartments (leaves, roots, heartwood and
sapwood), the autotrophic respiration, the production of litter, the plant mortality and the
degradation of soil organic matter (CENTURY model; Parton et al., 1988). The hydrological
processes for the soil reservoir rely on a double bucket scheme (Ducoudré et al., 1993). The
link between the water and carbon modules is via photosynthesis, which is based on the leaf-
scale equations of Farquhar et al., (1980) for C3 plants, and Collatz et al. (1992) for C4 plants,
that are then integrated over the canopy by assuming an exponential attenuation of light. The
FAPAR by each layer of the canopy is calculated from the leaf area index (LAI) following a
Beer-Lambert extinction law (Bacour et al., 2015).
ORCHIDEE uses the concept of the plant functional type (PFT) to describe the vegetation
distribution, with 13 PFTs (including bare soil) that can co-exist in each grid cell. Except for
the phenology (see a recent description in MacBean et al., 2015), the equations governing the
different processes are generic, but with specific parameter values for each PFT. Detailed
descriptions of model equations can be found in numerous publications (see for instance
Krinner et al., 2005). ORCHIDEE can be run at either global scale on a grid, or at site-level
using point-scale surface meteorological forcing variables. It is the land surface component of
the Institut Pierre Simon Laplace (IPSL) Earth System Model, and the version that we used
corresponds to CMIP5 simulations in the IPCC 5[th] Assessment Report (Dufresne et al., 2013).
However, in this study the model is run offline using the ERA-Interim 3-hourly near surface
meteorological forcing fields (Dee et al., 2011) aggregated at the spatial resolution of the
atmospheric transport model for the global simulations (see § 2.3.2). However, when we
assimilate in situ flux data in the second step, we force the model with the gap-filled half-
hourly meteorological data measured at each site. The global PFT map was derived from the
high-resolution IGBP AVHRR land data set (Vérant et al., 2004). The carbon pools are
brought to equilibrium (spin-up procedure) for both site and global scale simulations by
cycling the available meteorological forcing over several millennia, to ensure that the long-



term net carbon flux is close to zero. For the global simulation in third step, we spun-up the
model recycling the 1989-1998 meteorology and then used a transient simulation from 1990
to 2001 with changing climate (ERA-Interim) and increasing $CO_2$, before starting the
optimization with atmospheric data over 2002-2004. For the site simulations (i.e., the
assimilation of flux data) we recycled the available in situ meteorological forcing to spin-up
the model, with present day $CO_2$.

### 2.3.2 LMDz model

The transport model used in this study is version 3 of the General Circulation Model (GCM),
LMDz (Hourdin and Armengaud, 1999) with a horizontal resolution of 3.75° (longitude) x
2.5° (latitude) and 19 sigma-pressure layers up to 3 hPa. The calculated winds (u and v) are
relaxed to the ECMWF ERA-40 meteorological data (Uppala et al. 2005) with a relaxation
time of 2.5h (guiding) in order to realistically account for large-scale advection (Hourdin et
al., 2000). Deep convection is parameterized according to the scheme of Tiedtke (1989) and
the turbulent mixing in the planetary boundary layer is based on a local second-order closure
formalism. The LMDz GCM model has been widely used to model climate (IPCC, 2007,
2013) and its derived transport model has been used for the simulation of chemistry of gas
and particles and greenhouse gases distributions (Hauglustaine et al., 2004; Folberth et al.,
2005; Bousquet et al. 2005, 2006; Rivier et al., 2006). For this study, we used pre-calculated
transport fields, as described in Peylin et al. (2005), that correspond to the sensitivity of
concentration at each atmospheric site and each month to the surface flux of each model grid-
cell for each day (often called influence functions). The sensitivities (using inter-annual
winds) were calculated with the "retro-transport" formulation implemented in the LMDz
transport model (Hourdin et al. 2006). This approach decreases the computing time of the
optimization compared to the use of the full forward LMDz model at each iteration, as the
transport is replaced by a matrix multiplication with the vector of surface fluxes. Note that the
initial 3D state of the atmospheric concentrations was be defined from Chevallier et al. (2010)

### 2.3.3 Parameters optimized

The optimized parameters are described in Table 1, and their prior values, uncertainty and
range are given in Table 2. In the most recent studies using ORCHIDAS at site scales a large
set of ORCHIDEE parameters has been optimized (Kuppel et al., 2014; Santaren et al., 2014;
Bacour et al., 2015). In this study a smaller set was chosen, based on a Morris sensitivity





analysis (Morris, 1991; results not shown) that determines the sensitivity of the NEE and LE
to all model parameters at various FLUXNET sites (for each PFT), in order to reduce the
computational cost of the global optimization in step 3 (see §2.5). We considered 9 PFT-
dependent and 4 "global" (i.e. non PFT-dependent) parameters that control mostly the fast
carbon processes (diurnal to seasonal). In addition, we introduced a new parameter, $K_{soilC}$, to
scale the initial values (after spin-up) of the modeled slow and passive soil carbon pools, in
order to take account of all the historical effects not accounted for in the model that would
result in a disequilibrium of these pools in reality. For the site-specific optimizations with
FLUXNET data, we have one $K_{soilC,site}$ parameter per site. For the global scale optimization
step, we used 30 $K_{soilC,reg}$ parameters corresponding to 30 regions (see Fig. A2), thus the initial
soil carbon pools of all pixels within each region were scaled by the same value. The prior
value for all $K_{soilC}$ parameters was set to one, i.e. the default state of soil carbon pools is
assumed to be in equilibrium.
Overall (including all PFT-dependent parameters), we optimize 16 parameters related to
phenology, 36 to photosynthesis, 3 to respiration, 1 to the energy budget, 78 soil C pool
scalars (one for each FLUXNET site), and 30 regional soil C pool scalars for the global
simulations – a total of 184 parameters. Note that the soil C pool multipliers at the FLUXNET
sites are independent from the regional C pool multipliers, as the history of soil carbon over
large eco-regions of several millions square kilometers is rather heterogeneous (as it is mainly
related to previous land use changes), and most likely, the FLUXNET sites are not
representative of larger regions in terms of the soil carbon disequilibrium. The prior standard
deviation for each parameter is equal to 40% of the parameter range (lower and higher
boundaries) prescribed for each parameter following Kuppel et al. (2012). The parameter
ranges were specified following expert judgment of their meaning in the ORCHIDEE
equations and based on literature reviews or databases (such as TRY, Kattge et al., 2011).
**2.4   System description: a step-wise approach**
2.4.1 Stepwise assimilation of three data streams
The ORCHIDAS system relies on a stepwise assimilation of the three data streams described
in section 2.2. Figure 2 illustrates the flow of information in this sequential approach:
***Step 1 – Assimilation of MODIS-NDVI:*** Four parameters related to the seasonal cycle of the
vegetation (phenology) are optimized for the temperate and boreal deciduous PFTs (TeBD,



BoND, BoBD and NC3 – see caption of Table 2). These four deciduous PFTs alone are
considered in step 1 in this ORCHIDAS version because the tropical deciduous phenology
modules in ORCHIDEE require further modifications to improve the functions that control
leaf growth and fall in response to water availability (MacBean et al., 2015). Evergreen PFTs
were also not considered, as the there are no phenology modules related to these PFTs in the
model. The procedure is similar to that described in detail in MacBean et al. (2015) and
therefore only briefly recalled here. A simple linear relationship between the modeled
Fraction of Absorbed Photosynthetically Active Radiation (FAPAR) and MODIS-NDVI
observations is assumed, based on studies such as Knyazikhin et al. (1998). Following Bacour
et al. (2015), we use only the temporal information in the NDVI observations and not the
actual values, and thus we normalized both the model FAPAR output and the NDVI
observations to their $5^{th}$ and $95^{th}$ percentiles. The model was run for fifteen randomly selected
grid cells for each of the four PFTs using the ERA-Interim meteorological forcing. Only grid
cells that included vegetation fraction of greater than 60% for the PFT optimized were
considered. The fifteen sites for each PFT were included in one optimization for each PFT
following a multi-site approach, in which all observations are used simultaneously to optimize
the model parameters. The optimized parameters are described in Table 1. They correspond to
a scalar on the growing degree days (GDD) threshold for the start of the vegetation ($K_{pheno,crit}$),
a parameter controlling the use of carbohydrate reserve during the start of leaf growth
($K_{lai,happy}$), a temperature threshold for the onset of leaf senescence ($CT_{,senes}$) and the critical
age for leaves ($L_{agecrit}$).
***Step 2 – Assimilation of FLUXNET data:*** Mean daily NEE and LE flux measurements for 78
sites, including up to 10 years worth of data for each site, are used to optimize a set of model
parameters controlling the fast carbon and water processes (photosynthesis, respiration,
phenology – see Table 1). The site selection and the choice of a daily time step are described
in more details in Kuppel et al. (2014). These sites cover 7 of the PFTs in ORCHIDEE (see
Table 2). The posterior parameter values of the four phenology parameters derived in step 1,
and their associated uncertainties, are input as prior information in step 2. For the additional
parameters, the default ORCHIDEE values are used for the prior and the uncertainties are set
as described in §2.3.3. A multi-site optimization is performed for each PFT independently as
in step 1. Global parameters, i.e. those that are not PFT-dependent, were optimized for each
PFT and the mean across all PFTs was then calculated to define the prior parameter vector in
step 3 of the assimilation with atmospheric $CO_2$ data (at global scale). Such an approach was





chosen to allow us to optimize all PFTs in parallel and therefore to simplify the assimilation
process.
***Step 3 – Assimilation of atmospheric $CO_2$ concentrations:*** We use monthly mean $CO_2$
concentrations from 53 surface stations over three years (2002-2004) to provide a large-scale
constraint to the land surface fluxes (i.e. to match the global $CO_2$ growth rate, mean seasonal
cycle and its latitudinal variation, as well as the spatial gradients between stations). We use
the LMDz atmospheric transport model (see §2.3.2) to assimilate these observations. The set
of parameters optimized in step 2 are included in step 3, except for the albedo scaling
parameter ($K_{albedo,veg}$), as the net carbon fluxes are only weakly sensitive to that parameter. We
used the posterior parameter distributions from step 2 (parameter optimal values and
associated uncertainties) as prior information for step 3, and expanded the parameter vector to
include the 30 $K_{soilC}$ parameters that scale the initial soil carbon pools for large "spatially-
coherent regions" (see §2.1.2 and Fig. A2). The air-sea fluxes and fossil fuel and biomass
burning emissions are also accounted for (but not optimized) in this final step, in order to
close the global carbon budget within the atmospheric transport model (see §2.5).

## 2.4.2 Optimization procedure (for all steps):

In each step the statistically optimal parameter values are derived with an optimization
procedure following the principle of the 4-D variational assimilation systems (developed for
numerical weather prediction), using a tangent linear operator (and finite differences for a few
parameters, Bacour et al. 2015). Assuming that the errors associated with the parameters, the
observations and the model outputs follow Gaussian distributions, the optimal parameter set
corresponds to the minimum of a cost function, *J(x),* that measures the mismatch between i)
the observations (*y*) and the corresponding model outputs, *H(x),* (where *H* is the model
operator), and ii) the a priori (*$x_b$*) and optimized parameters (*x*), weighted by their error
covariance matrices (Tarantola, 1987; Eq. (1)):
$$J(x) = \frac{1}{2}\left[(H(x) - y)^T \, \mathbf{R}^{-1} \, (H(x) - y) \; + (x - x_b)^T \mathbf{B}^{-1} \, (x - x_b) \; \right] \tag{1}$$
**R** represents the error variance/covariance matrix associated with the observations and **B** the
parameter prior error variance/covariance matrix. At each step a different cost function is
defined with the observations and parameters related to that step (see Fig. 2). **R** includes the
errors on the measurements, the model structure and the meteorological forcing. Model errors
are rather difficult to assess and may be much larger than the measurement error itself.





Therefore we chose to focus on the structural error and defined the variances in **R** as the mean
squared difference between the prior model and the observations for both step 1 and step 2
(see Kuppel et al. 2013). For simplicity we assumed that the observation error covariances
were independent between the different observations and therefore we kept **R** diagonal (off-
diagonal terms set to zero), given the rapid decline of the model error auto-correlation beyond
one day (Kuppel et al., 2013). For step 3 we used a different approach, given the large bias in
the model a priori concentrations, and therefore followed the methodology of Peylin et al.
(2005) based on the observed and modeled temporal concentration variability at each site.
Overall, data uncertainties in the optimization procedure are between 0.1 and 0.45 for NDVI
(step 1), around 3-6 $gCm^{-2}d^{-1}$ for daily NEE, and 15-30 $Wm^{-2}$ for daily LE (step 2) and
between 0.1 ppm at remote oceanic stations and 4 ppm at continental sites (step 3).
The determination of the optimal parameter vector that minimizes $J(x)$ is performed by
successive calls to a "gradient-descent" minimization algorithm L-BFGS-B (Byrd et al.
1995), which is specifically dedicated to solving large nonlinear optimization problems that
are subject to simple bounds on the parameter values. In order to find the minimum of $J(x)$ the
algorithm requires the gradient of $J(x)$ (Jacobian) with respect to the ORCHIDEE parameters.
L-BFGS-B explores each parameter space simultaneously along the gradient of the cost
function, and uses an approximation of the Hessian (second derivative) of $J(x)$, which is
updated at each iteration, to define the size of the step at each iteration.
For step 1 and step 2, the model "$H$" simply corresponds to the land surface model: $H = S$,
with $S(x)$ representing the surface fluxes from the ORCHIDEE model using the parameter
vector, $x$. The gradients $dJ(x)/dx$ are calculated from the tangent linear model of ORCHIDEE
that was automatically generated by the numerical Transformation of Algorithms in Fortran
(www.fastopt.de), except for two parameters linked to the model phenology for which the
threshold functions prevent the use of a linear approximation. A finite difference approach
was used for these parameters.
For step 3, the model "$H$" corresponds to the composition of the land surface model with the
transport model: $H = T$ o $S$ (see Kaminski et al. (2002) for details), with $T$ representing the
LMDz transport model. $T$ is a linear operator for a non-reactive species: $T(S(x)) = \mathbf{T} \cdot S(x)$,
with **T** a matrix representation of the transport operator. It corresponds to the sensitivity of
$CO_2$ concentrations at each site and for each month to the daily surface flux of each model
grid-cell. It is then combined with the ORCHIDEE surface fluxes ($S(x)$) through a matrix





multiplication to derive $H(x)$. **T** has been pre-calculated for all atmospheric stations in order
to save computing time during the iterative optimization process (see §2.3.2). For simplicity
we use monthly mean values for both the fluxes $S(x)$ and the transport sensitivities (**T**) in the
computation of the gradients $dJ(x)/dx$.
For improved minimization efficiency, the inversion is preconditioned (following Chevallier
et al., 2005), which means that L-BFGS-B is fed with the control variable $x' = B^{-1/2}(x -$
$x_b)$, rather than with $x$, as this homogenizes the range of variation of the optimized
parameters.

### 2.4.3 Error estimation

The posterior parameter error covariance matrix, **A**, can be approximated to the   inverse
Hessian of the cost function, using the linearity assumption at the minimum of $J(x)$. It can be
derived with the Jacobian of the model at the end of the minimization (i.e. the last iteration),
$\mathbf{H}_\infty$, following Tarantola (1987):

$$\mathbf{A} = [\mathbf{H}_\infty^{T}.\mathbf{R}^{-1}.\mathbf{H}_\infty + \mathbf{B}^{-1}]^{-1}$$

(4)

Note that for step 3, $\mathbf{H}_\infty = \mathbf{T}.\mathbf{S}_\infty$, where $\mathbf{S}_\infty$ is the Jacobian of the ORCHIDEE model at the
last iteration. The posterior parameter error covariance, **A**, can then be propagated into the
model state variable space (e.g. carbon fluxes and stocks), $\mathbf{A_{var}}$, given the following matrix
product (only used for the global fluxes in step 3):

$$\mathbf{A_{var}} = \mathbf{S}_\infty.\mathbf{A}.\mathbf{S}_\infty^{T}$$

(5)

The square root of the diagonal elements of $\mathbf{A_{var}}$ corresponds to the standard deviation, $\sigma$, of
carbon fluxes/stocks for each grid cell. In order to evaluate the knowledge improvement
brought by the assimilation, the uncertainty reduction between the prior ($\sigma_{prior}$) and posterior
($\sigma_{post}$) is determined as $1 - (\sigma_{post} / \sigma_{prior})$.

### 2.4.4 Additional processing steps

In order to analyze the fit to the atmospheric $CO_2$ concentrations in terms of the trend and
seasonal cycle, we decomposed the observed and modeled time series by fitting the monthly
mean values with a function comprising a first order polynomial term and four harmonics,
and then filtered the residuals of that function in frequency space using a low pass filter
(cutoff frequency of 65 days), following Thoning et al. (1989). The polynomial term defines
the trend while the seasonal cycle corresponds to the harmonics plus the filtered residuals.
The amplitude of the seasonal cycle is then calculated as the difference between the monthly
mean maximum and minimum for year 2003 (middle year of the optimization period).
Finally, we define the Carbon Uptake Period (CUP) as the sum of the days when the values of
the seasonal cycle extracted from the $CO_2$ concentration time series are negative (a negative
convention being for $CO_2$ removed from the atmosphere).

## 2.5 Prescribed emissions of carbon fluxes

In this section we describe the other components of the carbon cycle (apart from the surface C
exchange with terrestrial vegetation) that are imposed in step 3 of the optimization process as
fixed fluxes.

### 2.5.1 Ocean fluxes

The ocean contributes to an uptake of about a quarter to a third of the anthropogenic
emissions with significant year-to-year variations (Sabine et al., 2004). For this version of the
ORCHIDAS, we developed a statistical model to estimate the spatial and temporal variations
(monthly) of the ocean surface $CO_2$ partial pressure ($pCO_2^{SW}$), and from that the air-sea $CO_2$
fluxes, using satellite and in-situ ocean measurements and model outputs. The air-sea $CO_2$
fluxes are primarily controlled by the ocean biogeochemistry, the horizontal transport and the
vertical mixing in the ocean and the atmospheric forcing ($CO_2$ partial pressure at the interface
to the water ($pCO_2^{ATM}$) and wind); they can be defined from the following equation:
$$F_{CO2} = K_{ex} \times (pCO_2^{SW} - pCO_2^{ATM}) \qquad\qquad (6)$$
where $K_{ex}$ stands for the exchange coefficient and $F_{CO2}$ the $CO_2$ flux from the sea surface
water to the atmosphere.
The computation of $pCO_2^{SW}$ is performed using feedforward artificial neural networks, i.e., a
MultiLayer Perceptron (MLP; Rosenblatt 1958) that maps a set of spatio-temporal variables
(input) onto observed $pCO_2^{SW}$ data. We use a two-step approach: the first step to derive a
monthly mean $pCO2^{SW}$ climatology and the second step to correct for the year to year
variations. The $pCO_2^{SW}$ observations come from the Global Surface pCO2 (Lamont-Doherty
Earth Observatory, LDEO) Database (Takahashi et al., 2009). The inputs are a series of
variables connected to the spatial and temporal evolution of $pCO_2^{SW}$: i) sea surface
temperature (SST), sea surface salinity (SSS) and mixed layer depth (MLD) as a proxy of the





physical processes (these fields come from a re-analysis of the NEMO-OPA ocean model
(Madec et al., 1998) with the assimilation of several satellite observations), ii) chlorophyll
content from SeaWiFS, as a proxy of the biogeochemistry (CHL), iii) spatial and temporal
coordinates (LAT, LON and MONTH) and the $pCO_2^{SW}$ at previous time step (recursive
approach), i.e.:
$$\{pCO_2^{SW}\}_m = MLP\left(\{SST, SSS, MLD, CHL\}_{(m-2,m-1,m)}, \{pCO_2^{SW}\}_{(m-2,m-1)} \; LAT, LON\right) \quad (7)$$
with $m$ the monthly index. The available data (20685 points) is divided into two parts: 75% is
used for the learning phase of the ANN and 25% for the validation phase. The overall
performance of the neural network for extrapolating the spatial and seasonal distribution of
$pCO_2^{SW}$ is relatively good, with a spatio-temporal correlation coefficient between the
estimated $pCO_2^{SW}$ and the independent observations of 0.80.
$pCO_2^{ATM}$ at the surface are taken from a global simulation of atmospheric $CO_2$ concentrations
with optimized fluxes (Chevallier et al. 2010). $K_{ex}$ is defined as the product of $k$, the gas
transfer velocity, taken from the Wanninkhof (1992) formulation using winds from ERA-
Interim, and $s$, the solubility of $CO_2$, taken from the Weiss formulation (Weiss, 1974). The
system is further described in Roedenbeck et al. (2015). The global ocean sink is around 1.60
$PgC.yr^{-1}$ for the period 2002-2004 used in step 3. It is within the uncertainty range of the
Global Carbon Project estimates (Le Quéré et al., 2015) if we account for the pre-industrial
ocean out-gazing flux included in our "delta $pCO_2$" approach. Its temporal evolution is
depicted in Fig. A1
## 2.5.2 Global fossil fuel and cement $CO_2$ emissions
We have used a recently developed $CO_2$ fossil fuel and cement emission product (see
[http://www.carbones.eu/wcmqs/](http://www.carbones.eu/wcmqs/)) that covers the period 1980 to 2009 at the spatial resolution
of 1° x 1° and hourly resolution. It is based on EDGAR v4.2 spatially distributed annual
emissions (Olivier et al., 2012) and time profiles developed by the University of Stuttgart. It
was assumed that EDGAR delivers the most up-to-date spatially distributed and sector
specific emissions, based on national emission statistics. IER further applied country and
sector specific time profiles, taking into account monthly, daily, and hourly variations
depending on the sector. The derivation of the time profiles relies on different data sets (e.g.
Eurosstat, ENSTO-E, UN monthly bulletin) as well as correlations between recorded
emissions and climate variables. Currently, the temporal profiles are derived mostly from data





sets over Europe that were extrapolated using information on climate zone, average monthly
temperature for the seasonal cycles and similarity in socio-economic parameters like
population and Gross Domestic Product (GDP). The annual mean emission for the period
2002-2004 is 7.14 PgC.yr$^{-1}$.
### 2.5.3 Fire emissions:
Fire      emissions      data      from      the      Global      Fire      Data      (GFEDv3      –
http://www.globalfiredata.org/Data/index.html) are prescribed in the ORCHIDAS. The
GFEDv3 data are broken-down into 6 sectors (deforestation, peat fires, savanna fires,
agriculture, forest fires, and woodland) that are further grouped into 3 main types. We
generated fluxes of $CO_2$ relevant for typical "burning - regrowth" processes, as detailed in
Appendix A2. The first type corresponds to deforestation and peat fires with carbon
permanently lost to the atmosphere, the second to agriculture and savannah fires which are
assumed to be compensated by a sink during the regrowth period (i.e. with zero annual net
emission for each pixel) and the third to woodland and burnt forests which are assumed to be
at steady state for a given region (10 sub-continental scale regions) over the period covered by
GFEDv2 (i.e. regrowth of nearby forest compensates for the burned forest derived in GFED).
The sum of these three components leads to the global flux, with a gross emission around 2.1
PgC.yr$^{-1}$ and a net emission after regrowth of only 1.1 PgC.yr$^{-1}$ (Fig. A2 in Appendix) that is
prescribed to the ORCHIDAS over the period 2002-2004.
**3   Results**
**3.1 Model fit to the data**
### 3.1.1 Step 1: assimilation of MODIS NDVI data
The optimization in Step 1 resulted in an improved fit to the MODIS NDVI observations for
the four PFTs considered (TeBD, BoND, BoBD, NC3, see §2.4) as seen in Fig. 4, which
shows the mean seasonal cycle across the 2000-2008 period for all sites for each PFT. The
most prominent change after the optimization was a substantially shorter growing season for
all PFTs due to an earlier start of leaf senescence. This was caused by both a lower critical
leaf age ($L_{agecrit}$) and a higher temperature threshold for senescence ($CT_{senes}$) (Fig. 8). The
impact on the start of leaf growth was less dramatic but important nonetheless, with a shift to



a later start of leaf growth as a result of an increase in the $K_{pheno,crit}$ parameter which acts as a
scalar on the threshold of Growing Degree Days (GDD) used to trigger leaf onset (see
Appendix A in MacBean et al., 2015). Overall, a mean reduction in RMSE of 23, 17, 58 and
19% was achieved for TeBD, BoBD, BoND trees and NC3 grasses respectively, with the
greatest improvement for BoND trees. The mean correlation between the normalized MODIS-
NDVI and modeled FAPAR time series over the 2000 – 2008 period increased for TeBD and
BoND trees and NC3 grasses (prior and posterior of 0.9 to 0.93, 0.42 to 0.91 and 0.6 to 0.66,
respectively). The prior correlation of 0.55 remained similar after the assimilation for BoBD
trees.
Following the improvement at the sites selected for the optimization, we evaluated the impact
for each PFT at the global scale using the global median correlation between the MODIS-
NDVI and the model FAPAR time series (from all pixels where the fraction of a given PFT is
above 60%, see Maignan et al. 2011). The global correlation increased for BoND trees and
NC3 grasses from 0.36 to 0.91 and 0.53 to 0.59 (prior to posterior), respectively. It remains
stable for BoBD (0.54) or slightly increased for TeBD (0.88 to 0.89).
### 3.1.2 Step 2: assimilation of FLUXNET data
The optimization in Step 2 brings an improvement to the simulated NEE and LE for all seven
PFTs considered, with Fig. 5 showing the corresponding PFT-averaged mean NEE seasonal
cycles (mean across all sites/years). NEE is overestimated by the prior model for all PFTs on
average. This is partly due to the model spin-up procedure, which brings each simulated site
to a near equilibrium state with a mean NEE close to zero (i.e. no net carbon sink, see §2.1.1).
This bias is significantly corrected by the optimization to match the observed carbon uptake at
most sites, notably via the scaling of the initial soil carbon pool content at each site
(parameters $K_{soilC,site}$; Table 1) which thus significantly reduces the ecosystem respiration
(Kuppel et al., 2014). Overall, the largest reductions of model-data RMSE are found in
temperate forests (TeNE, TeBE and TeBD), where the RMSE decreased by more than 25%
compared with the prior model. The improvements are less significant for the other PFTs,
with RMSE reductions between 10 and 18%.
In addition, the optimization increases the NEE seasonal amplitude in temperate evergreen
forests (TeNE and TeBE) and temperate broadleaf deciduous forests (TeBD), and reduces the
amplitude for boreal needle leaf forest (BoNE) and natural C3 grasses (NC3), in agreement



with the observations (except for BoNE where the amplitude decrease is too large). Despite
the better model-data agreement for evergreen broadleaf forests (TrBE and TeBE), the
optimized model still fails to catch some seasonal features such as a persistent carbon uptake
(i.e. negative NEE) in the dry season for the tropical regions (TrBE) and nearly-null carbon
exchange in the first months of the year for temperate regions (TeBE). These results are
discussed further in Kuppel et al. (2014), who used a similar optimization set-up with a
slightly different parameter set – see §2.3.3. Similar improvements, although of smaller
amplitude, occur for the latent heat fluxes (not shown).

### 9    3.1.3 Step 3: assimilation of atmospheric $CO_2$ data

The final optimization step with the atmospheric $CO_2$ concentrations provides a large
improvement of the fit to the observed concentrations at most stations. The cost function $J$
was reduced through the minimization by a factor of 5.7 within 37 iterations.
Figure 6 illustrates the simulated concentrations for four stations (representative of different
conditions) with the standard prior parameter vector (used in step 1), the posterior vector from
step 2 (used as prior in step 3) and the posterior vector from this last step. The improvement
in the fit to the observations can be quantified with the reduction in RMSE (from the prior to
the posterior of step 3) - the largest reduction is at the South Pole station (73%) and is on
average around 25% across all sites. Note that for a few stations the fit is slightly degraded
(up to 10%) except for one Pacific site (regular ship measurements around the equator,
POCN00) for which there is a 40% degradation, possibly due to small biases in the simulation
of the ITCZ position in LMDz. When calculated with respect to the standard prior (used in
step 1) the RMSE decrease is slightly larger on average, especially for the northern mid to
high latitude stations. For these stations the optimization performed in step 2 with FLUXNET
data led to a significant improvement of the mean seasonal cycle amplitude of the
atmospheric $CO_2$ data, as discussed in Kuppel et al. (2014).
We then investigated the fit to the observed $CO_2$ concentrations in terms of the mean seasonal
cycle and trend (see section 2.4.4). With only three years of data the mean trend is more
difficult to define as it varies between stations; however, the optimization in step 3 increases
the net land carbon sink in order to match the observed trend at most stations. If we take the
Mauna Loa and South Pole stations that are representative of an integration of the fluxes at
hemispheric scales, the prior $CO_2$ trend of 2.8 and 2.9 ppm.yr$^{-1}$ respectively, is reduced to 2.1





and 2.2 ppm.yr$^{-1}$ close to the observations (2.1 ppm.yr$^{-1}$ for both). The left panel of Fig. 7
illustrates changes in the amplitude of the simulated seasonal cycle at each station (see
definition in §2.4.4). The values correspond to relative changes between the prior and
posterior of the absolute difference between observed and modeled amplitude ($\left[\left|\Delta A_{poste}\right| - \right.$
$\left. \left|\Delta A_{prior}\right|\right]/\left|\Delta A_{prior}\right|$). They reveal an improvement in the seasonal cycle amplitude at nearly
all stations of the southern hemisphere ($\approx$ 40% improvement) and at the majority of the
northern hemisphere stations ($\approx$ 15%). A few stations in north East Asia (3) and northwest
America (4) show a small degradation of the amplitude ($\approx$ 15%). The right panel of Fig. 7
displays the changes of the Carbon Uptake Period (CUP, see §2.4.4) expressed in terms of
relative changes between prior and posterior of the absolute values of model-data differences,
as for the amplitude. Most stations reveal an improvement of the CUP of around 20%, which
is slightly lower than the improvement for the seasonal cycle amplitude.

### 3.2 Consistency of the step-wise optimization

The main issue with a step-wise data assimilation system (versus a simultaneous approach)
concerns the potential degradation of the model – data fit for the different data streams that
are assimilated in previous steps. We noted that $CO_2$ concentrations were already improved
when NDVI and FLUXNET data are assimilated (see §3.1.3), but we need to check if the
final parameter set from step 3 leads to a degradation of the fit to MODIS-NDVI (step 1) and
to FLUXNET (step 2) data compared to the fit achieved during the respective steps and, in the
case of a significant degradation, if we still have an improvement for these data streams
compared to the initial *a priori* fit.
Figure 8 summarizes the performance of the model data fit for MODIS-NDVI and
FLUXNET-NEE data streams for the prior and posterior of each step by evaluating the
median RMSE between the model and the observations across all sites. The values are
calculated for each PFT separately. In this section, we keep in mind the fact that we do not
optimize the same PFTs with FLUXNET data and with MODIS-NDVI.

### Consistency for MODIS-NDVI

First, we notice again the significant RMSE reduction between the prior and step 1, as
discussed in section 3.1. The fit to MODIS-NDVI (normalized data) for step 2 and step 3
shows only a significant degradation (increased RMSE) for temperate broadleaf deciduous





forest (TeBD), which decreases the improvement achieved in step 1 (compared to the prior)
by a factor of two. A marginal degradation for natural C3 grassland (NC3) is obtained after
step 3: the RMSE increases slightly from 0.24 to 0.26, but is still lower than the prior value of
0.3. There is no degradation for boreal needleleaf deciduous trees (BoND), but a surprising
small decrease of the RMSE (i.e. improvement in the model-data fit) for boreal broadleaf
deciduous forests (BoBD; from 0.26 to 0.23). In this latter case, the use of additional
parameters in steps 2 and 3 (see §2.4) allows further improvement of the fit between the
normalized FAPAR and NDVI time series. On average the degradation of the fit to NDVI is
thus very limited in step 2 and step 3, and in no case is the RMSE worse than the prior.
Consistency for FLUXNET data
Figure 8 again reveals the significant reduction of the RMSEs for NEE in step 2 compared to
the standard prior or to the posterior of step 1 for most PFTs, except BoNE. We see only
small degradations (increases) in RMSE between step 2 and step 3 for temperate needle leaf
evergreen forests (TeNE: from 1.06 to 1.13 $gC.m^2.d^{-1}$), temperate broadleaf evergreen forests
(TeBE: from 1.06 to 1.09 $gC.m^2.d^{-1}$), temperate broadleaf deciduous forests (TeBD: from 1.06
to 1.13 $gC.m^2.d^{-1}$) and boreal needle leaf evergreen forests (BoNE: from 0.59 to 0.60 $gC.m^2.d^{-1}$
$^{1}$). An interesting feature to notice is that the NEE RMSE increases between the prior to the
posterior of step 1 (i.e. before NEE has been used in the optimization in step 2). Using remote
sensing products of vegetation activity or "greenness" (e.g. NDVI) to calibrate the phenology
of ORCHIDEE thus does not always improve the simulated NEE, the possible reasons for
which were discussed in Bacour et al. (2015) who used the same LSM and assimilation
system. Overall, the reduction of the improvement of the model data fit to the NEE (step 3
versus step 2) is marginal (limited to a few percent), thus further suggesting the consistency of
our step-wise approach. Similar results are also obtained for the latent heat flux (LE) (not
shown).
**3.3 Estimated parameter values**
We now discuss the parameter values, focusing on the changes obtained though the
successive steps. Figure 9 presents the prior and posterior values for each parameter together
with their associated uncertainties (estimated through Eq. (4)) and the allowed range of
variation. Note that nine parameters are PFT-dependent while four are global (non PFT-
dependent). For the global non PFT-dependent parameters included in the step 2 optimization,





we took the mean value (see §2.4) as the prior for step 3. Note finally that the parameters
linked to the initial soil carbon pools ($K_{soilC,site}$, $K_{soilC,reg}$) are not shown in Fig. 9 as they are
too numerous (though see Fig. A2 for the regional values).
If we first consider the phenology parameters optimized in step 1 ($K_{lai,happy}$, $K_{pheno,crit}$, $L_{age\_crit}$,
$C_{T,senes}$; see Table 1) we see that for most PFTs they do not change significantly between step
1 and step 3, although they differ significantly from the prior. There are few exceptions,
including $K_{pheno,crit}$ (the threshold for the start of the growing season) for Boreal Needleleaf
deciduous forests and $K_{lai,happy}$ (level of carbohydrate use) for temperate and boreal broadleaf
deciduous forests (TeBD, BoBD). Note that a few phenology parameters hit one of the
physical bounds, which may indicate model structural errors or model parameter equifinality.
For most phenology parameters, the uncertainties are strongly reduced during their first
optimization (step 1), except for a few cases like $C_{T,senes}$ for C3 grassland. Note finally that a
more in depth spatio-temporal validation demonstrated the generality of the optimized
phenology parameters across multiple sites (for further details see MacBean et al., 2015).
For the photosynthesis parameters ($V_{cmax}$, $G_{s,slope}$, $C_{Topt}$, $SLA$, $f_{stress}$; see Table 1), we find a
similar result with little changes between step 2 and step 3, but still a significant departure
from the prior values. Most parameters are well constrained by the inversion, with posterior
uncertainties that are greatly reduced compared to the prior, except for Tropical broadleaf
rain-green forest (TrBR) and Boreal needle-leaf deciduous forest (BoND) for which there is
nearly no constraint on $G_{s,slope}$, and $f_{stress}$ (see Table 1).
The non-PFT dependent respiration-related parameters ($HR_{H,c}$, $Q_{10}$, $MR_b$) mostly change in
step 2 and only slightly in step 3 (with an additional reduction of the error) in order to fit the
large-scale constraint provided by the atmospheric observations. The values of the scalar of
the initial soil carbon pools size for the FLUXNET site optimizations ($K_{soilC,site}$, one parameter
per site, not shown) were largely reduced on average, in order to decrease the heterotrophic
respiration (see Kuppel et al. (2014) for additional discussion). In step 3 the same scalars that
were defined for an ensemble of large regions ($K_{soilC,reg}$) have decreased in the southern
hemisphere (less than 10%; see Fig. A2 in Appendix A3) and slightly increased in the
northern hemisphere (around 1%), to achieve a better match to the atmospheric $CO_2$ growth
rate and north-south gradient. Importantly, we notice that for step 3, the fit to the atmospheric
$CO_2$ concentrations (especially to the trend) is achieved mainly by small changes in $K_{soilC,reg}$
and in few other respiration-related parameters. Note finally that the parameter controlling the





albedo ($K_{albedo,veg}$), modified with the FLUXNET observations only (see §2.4), is not well
constrained by the optimization (only a small reduction in uncertainty). Overall, most
parameters appear to be well constrained when first optimized, with only small changes in the
following steps. This suggests that the targeting of different parameter subspaces in the
various optimisation steps was well-chosen.
**3.4 Estimated carbon fluxes and uncertainties**
The main objective of a carbon cycle data assimilation procedure is to improve the simulated
land surface net and gross carbon fluxes as well as the simulated carbon stocks for both
present and future conditions. Given the focus of the paper, i.e. to describe the potential of a
step-wise global carbon cycle data assimilation system, we only discuss a few large-scale
features of the optimized annual net and gross carbon fluxes, as well as one of the carbon
stock variables (forest above-ground biomass). We thus do not discuss the inter-annual flux
variability.
Large-scale annual mean net fluxes
The mean annual carbon fluxes (NEE) for the globe, northern extra tropics, tropics, and
southern extra tropics are reported in Fig. 10 for the 2000-2009 decade for the prior and
posterior model simulations for all steps together with one other estimate of the land surface
residual from the Global Carbon Project (GCP, Le Quéré et al, 2015) over the same decade.
The prior NEE indicates a total sink of 0.5 PgC.yr$^{-1}$ over this period, from both the northern
and tropical regions. Such a prior sink is due to the increase of atmospheric $CO_2$ during the
transient simulation following the spin-up (1990-2009, see section 2.3.1) and climate
variability. Changes from the prior are rather small in step 1 (assimilation of MODIS NDVI))
with an increase of the northern sink by 0.12 PgC.yr$^{-1}$ and a decrease of the tropical sink by
0.05 PgC.yr$^{-1}$ (Fig. 10). Step 2 (assimilation of FLUXNET data) does not significantly change
the net C sink from step 1, with only a small increase in the tropical sink by 0.1 PgC.yr$^{-1}$. The
assimilation of atmospheric $CO_2$ data in step 3 provides a large-scale constraint, as already
discussed, and increases the total land sink to 2.2 PgC.yr$^{-1}$, a value in much closer agreement
with the estimate by the GCP. A larger tropical NEE uptake is responsible for the large
increase of the terrestrial biosphere C sink (from 0.3 PgC.yr$^{-1}$ in step 2 to 1.7 PgC.yr$^{-1}$) while
the sink in the north increases by less than 0.1 PgC.yr$^{-1}$. The comparison with the GCP
number should be taken with caution. The ORCHIDAS estimated sink include all effects



(natural and anthropogenic), since that we used atmospheric $CO_2$ as a global constraint. Thus
the optimized parameters must account for any missing processes like nitrogen limitation or a
proper description of agricultural processes and management. However, the GCP number is
only for the anthropogenic uptake, excluding the pre-industrial sink due for instance to river
export of carbon (around 0.45 PgC.yr$^{-1}$; Regnier et al. 2013).
Spatial distribution of the annual mean net flux
Figure 11 shows the spatial distribution of NEE averaged over 2002-2004 for the standard
prior and posterior after step 3. The large tropical net land carbon sink that is inferred in step
3 is mainly explained by an increase of the carbon uptake for the tropical forests of the
Amazon basin and equatorial Africa, as well as a decrease of the carbon release on the
southern edge of the Amazon basin (tropical rain-green forests and grasses). In the northern
mid-high latitudes only smaller regional changes from the prior occur. For Europe, most of
north Asia and Canada, the strength of the C sink slightly decreased from the prior (up to 30
gC.m$^2$.yr$^{-1}$), while for central USA the strength of C source slightly decreased. If we now
consider the uncertainties on the net annual carbon flux that arise from the parameter
uncertainty (second row of Fig. 10; Eq. (5)) we observe a very large reduction (compared to
the prior) in the monthly flux uncertainty (averaged over the three years used in step 3) over
tropical forests. It is reduced by a factor four with initial values around 150 gC.m$^2$.y$^{-1}$ and
posterior values between 22 and 66 gC.m$^2$.y$^{-1}$. For mid-to-high latitude boreal ecosystems, the
uncertainty reduction is smaller, but the posterior errors are slightly lower than over the
tropics, between 18 and 55 gC.m$^2$.y$^{-1}$.
Large-scale annual mean Gross Primary Production (GPP)
For the GPP the relative changes from the prior are smaller than for the NEE (Fig. 10b). The
mean annual total GPP is 169, 160, 154 and 156 PgC.yr$^{-1}$ for the prior and posterior of step 1,
2 and 3, respectively. The small overall decrease (8%) brings the GPP slightly closer to the
estimate by Jung et al. (2011), around 120 PgC.yr$^{-1}$, based on a statistical Model Tree
Ensemble (MTE) that upscaled the in-situ flux measurements (resulting from the partition of
measured NEE into GPP and total ecosystem respiration). The decrease in GPP occurs mainly
in the northern hemisphere after step 1 (-10 PgC.yr$^{-1}$) following the decrease in $V_{cmax}$ (Fig. 9)
while it remains relatively stable over the tropics across all steps. Note that i) the study of
Welp et al. (2011) suggests a GPP around 150 PgC.yr$^{-1}$, similar to our estimate, based on



measurements of $^{18}O/^{16}O$ ratio in atmospheric $CO_2$ and ii) Koffi et al. (2012) found optimized
GPP of 146 PgC.yr$^{-1}$ from a CCDAS using the BETHY model.

## Above-ground forest biomass

We analyze the impact of the optimization on the forest above-ground biomass at equilibrium
(i.e. after spin-up; see Fig. 12) as an example of the impact on model C stocks, and compare
the simulated values, for the same three latitude bands than above, to the estimate based on
field observations and remote sensing data. This product, which was produced in the
GEOCARBON project (and thus is referred to by the same name), integrates a pan-tropical
biomass map (Avitabile et al., 2016) with a boreal forest biomass product (Santoro et al.,

10 2015).

For the northern extra tropics, the prior above-ground C stock (~180 PgC) is reduced by the
optimization to 140 PgC, mainly through the decrease of the growing season length in step 1
with the assimilation of MODIS-NDVI. The significant decrease in GPP during step 1 (18 %)
led indeed to a similar decrease of the forest biomass (16%). Parameter changes through the
assimilation of FLUXNET and $CO_2$ data have a smaller impact (a change of less than 5 PgC).
These changes in the northern extra tropics bring the estimates by the ORCHIDEE model
closer to the satellite-based GEOCARBON product (~ 120 PgC).
For the tropics, while there is nearly no change with the assimilation of MODIS-NDVI in step
1, the use of FLUXNET data leads to a significant increase of the forest above ground
biomass (close to 25%). Such an increase does not correspond to an increase of the GPP (Fig.
10) but to changes in the autotrophic respiration parameter ($MR_b$) that lead to a decrease of
autotrophic respiration and an increase of NPP. The value does not change through step 3 and
remains significantly higher than the data-driven estimate. Note however that the lower value
in the GEOCARBON product could be partly due to the fact that we did not yet account for
land use effects in the CCDAS, such as deforestation in the Amazon.

## 4 Discussion and conclusions

In this paper we have described a first global Carbon Cycle Data Assimilation System that
assimilates three major carbon-cycle data streams, namely MODIS-NDVI observations of
vegetation activity at 60 sites, FLUXNET NEE and LE measurements at more than 70 sites,
and atmospheric $CO_2$ concentrations at 53 surface stations over three years in order to





optimize the C cycle parameters of the ORCHIDEE process-based LSM (ORCHIDEE-
CCDAS). The study details the concept, the implementation and the main results of a
stepwise assimilation approach where the data streams have been assimilated in three
successive steps (including a propagation of the retrieved posterior parameter distributions
from one step to the next).
The assimilation of MODIS-NDVI (60 grid cell points, step 1) improved the phenology of
ORCHIDEE with a significant reduction of the growing season length and thus a direct
impact on the GPP. The results are similar to those presented in MacBean et al. (2015) who
describe the impact of such optimization on the global FAPAR simulations and the
improvement in the bias of the calculated leaf onset and senescence dates in more detail. The
optimization with FLUXNET data (78 sites, step 2) led to large improvements in the seasonal
cycle of the NEE and LE fluxes, constraining primarily the photosynthetic processes. Some
discrepancies remain due to site heterogeneity (i.e. different species and edaphic conditions)
that the model does not fully capture, and due to missing processes in the model (see Kuppel
et al. (2014) for a more thorough discussion). However, without the assimilation of
atmospheric $CO_2$ concentrations, the global (and continental) net carbon balance after step 2
was still clearly outside the admitted range (as reported by the GCP in Le Quéré et al. (2015),
which highlights the importance of assimilating a data stream such as this that provides
information at larger scales (constraining large scale respiration fluxes). The use of
atmospheric $CO_2$ concentration as an overall constraint in step 3 was technically challenging
as it required the coupling of ORCHIDEE with an atmospheric transport model in forward
and reverse mode (i.e. to compute the cost function and its gradients at each step of the
minimization process). As a result of the final step, we were able to fit the atmospheric $CO_2$
growth rate and thus to derive a land C sink compatible with current best estimates from the
GCP. The assimilation of $CO_2$ data also slightly changed the seasonality of the NEE, which
improved the fit to the atmospheric $CO_2$ seasonal cycle. Note that assimilating only $CO_2$ data
would lead to a similar global land C sink but with a different model parameter set not
compatible with the information provided by MODIS-NDVI and FLUXNET data.
The consistency of the stepwise approach has been evaluated with back-compatibility checks
after the final step (step 3: assimilation of atmospheric $CO_2$ concentration). The optimized
model with the final set of parameters does not degrade the fit to MODIS-NDVI and
FLUXNET data that were assimilated in the first two steps (only minor changes of the





RMSEs occur; see Fig. 8). This result has two important consequences. Most importantly it
suggests that current state of the art LSMs (at least ORCHIDEE) have reached a level of
development where consistent assimilation of multiple data streams is finally possible. This
overcomes the most important limitation noted by Rayner (2010) to the widespread use of
CCDAS systems. At a more technical level it suggests that stepwise assimilation is a valid
and feasible approach. Although we only carried the estimated parameter uncertainties from
one step to the next (as a first simple approach), and not the full error variance-covariance
matrix, we were able to propagate enough information to maintain an optimal model-data fit
after the last step for the three data streams (see MacBean et al. (2016) for a more specific
analysis of this issue). However, not propagating the covariance terms may have a larger
impact for the reduction of the inferred parameter uncertainties (see for instance the large
parameter / flux error reduction in Fig. 9 / Fig. 11). The order of the different steps was
dictated by the number of parameters we choose to expose to each data stream, from only a
few phenology parameters for NDVI up to the largest set for atmospheric $CO_2$. Recall that
under the fundamental theory the order of assimilation is unimportant. Testing whether our
system meets this criterion is an important check on the robustness of the method but is not
technically feasible with the full-blown system; it is currently being tested with some smaller
models.
Most of the optimized parameter values have significantly changed compared to their prior
values, with a large error reduction for most (Fig. 9) that results in a strong constraint on the
simulated fluxes (Fig. 11). In the last step, the assimilation of atmospheric $CO_2$ data mainly
led to the optimization of respiration-related parameters, especially the regional soil carbon
multipliers ($K_{soilC,reg}$). Note that this was also the case for the BETHY-CCDAS, as described
in Rayner et al. (2005) (see their Table 2). This is linked to the difficult issue of representing
the effects of historical changes in land cover and land management as well as soil texture
effects on soil carbon dynamics, and the necessary choice of a standard spin-up procedure to
account for these effects. Ideally one would need to perform the optimization of the model
over a long historical period with LULCC and land management practices included and the
optimization of related parameters. However, this is not currently feasible at global scale and
uncertainties in the forcing would introduce as much difficulty as uncertainties in the initial
condition. The adjustment of the initial C pool contents is thus a logical compromise and
further investigations into the impact of the selected set-up (number of regions for $K_{soilC,reg}$,
their associated uncertainties) on the C fluxes simulated in the future are needed. Note that a

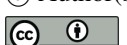



first improvement would be to include LULCC in the transient simulation (to define the initial
state) before the assimilation period.
Nonetheless, several limitations, inherent to the optimization of model parameters in a
CCDAS, need to be called to mind when evaluating these results (see also Rayner et al.,
2010). First, the structure of the land surface model (i.e. how biogeochemical processes are
represented) is critical. Any missing/misrepresented processes may have a direct impact and
thus lead to biases in the selected parameters. Note that this limitation could be even more
severe when using atmospheric $CO_2$ measurements, as these data provide a direct constraint
on the overall net C exchange between the atmosphere and the vegetation, thus including all
processes. As an example, the model sensitivity to atmospheric $CO_2$ increase (e.g. through the
parameters $V_{cmax}$ and $G_{s,slope}$) could be non optimal as the current model version does not
include explicit nitrogen and phosphorus limitations on photosynthesis. Second, the chosen
set of observations does not provide specific constraints on long term C processes such as tree
mortality, disturbance effects, or C allocation within a plant. For instance Fig. 12 illustrates
that the optimized model may still significantly overestimate tropical forest biomass. The
assimilation of above-ground biomass or soil carbon stock observations (i.e. site-level
measurements or regional estimates) should thus provide critical complementary information
(see Thum et al., in revision for AFM).
To conclude, this work is a step forward in terms of multiple data streams assimilation that
opens new perspectives for a better understanding of the carbon cycle and better predictions
of the fate of the land carbon sink in the 21[st] century as a consequence of anthropogenic
changes. As ORCHIDEE is part of the IPSL earth system model the impact of the
optimization on future climate change predictions will be assessed in a future study. However,
we first need to run the ORCHIDAS with a longer atmospheric $CO_2$ record (i.e. several
decades) in order to provide stronger constraints on parameters controlling the impact of
climate extremes on the net carbon fluxes at continental to global scales, and the sensitivity of
photosynthesis to increasing $CO_2$ concentration. The optimized model will allow a more in-
depth investigation of the trend and inter-annual variations of land surface C fluxes at
continental to regional scales, as well as their driving mechanisms. It will offer a more
reliable and robust process-based diagnostic of the land C cycle that is compatible with
current major data streams. Overall, we have illustrated the benefit of combining multiple
data streams in a process-based model to optimize different processes of the model, related to



different temporal and spatial scales. The optimization will be updated regularly as new
processes are integrated into the ORCHIDEE model, such as for instance land management
(Naudts et al., 2015).
**Code availability**
The ORCHIDEE model code and the run environment are open source
(http://forge.ipsl.jussieu.fr/orchidee) and the associated documentation can be found at
https://forge.ipsl.jussieu.fr/orchidee/wiki/Documentation. Note that the tangent linear version
of the ORCHIDEE model has been generated using commercial software (TAF;
http://www.fastopt.com/products/taf/taf.shtml). For this reason, only the "forward" version of
the ORCHIDEE model is available for sharing. The optimization scheme (in Python) is
available through a dedicated web site for data assimilation with ORCHIDEE
(http://orchidas.lsce.ipsl.fr/). Nevertheless readers interested in running ORCHIDEE are
encouraged to contact the corresponding author for full details and latest bug fixes. Finally,
the source code of the LMDZ atmospheric transport model can be found at
http://web.lmd.jussieu.fr/trac.
# Appendix
**A1. Ocean fluxes**
Figure A1 displays the air-sea fluxes from the statistical model.
**A2. Fire fluxes**
In order to account for fundamental differences between six fire flux categories provided by
the GFED product, we grouped these emissions into 3 types with specific treatments. The first
group includes C emissions from deforestation and peat fires, which are considered to be
permanent carbon lost to the atmosphere, at least over the considered time scales. These
fluxes are rescaled to an annual emission of 1.1 PgC.yr$^{-1}$ globally following typical values
reported in the literature for deforestation (Houghton R., 2003). The second group consists of
C emissions from agriculture and savannah fires, which are compensated by a C sink during
the regrowth of these biomes (i.e., savannah and some type of plants on the farmland). These





effects are not completely accounted for in ORCHIDEE as the model does not simulate
savannah and agriculture fire. Hence, the emissions over the whole period and for each pixel
become zero, but their seasonal variations are used. The final group includes emissions from
woodland and burnt forests. We considered that at steady state and for a given region certain
forests burn but that nearby forests are re-growing following older fires. We thus imposed
regrowth at the region scale given that the ORCHIDEE model version that we use does not
account for such regrowth. The main assumption is that over century time scale the
forest/woodland system is at steady state over a given region (few thousand square km), i.e.
there is no net deforestation. We selected an ensemble of small regions over which we
calculated the regrowth of these biomes. The derived emissions over the whole period and for
each region thus become zero; though we include their spatial and temporal variations. The
overall biomass burning flux considered in the CCDAS for the optimization process is the
sum of the three fluxes as described above.
**A3. Multipliers of the soil initial carbon pools**
Figure A2 provides the optimized values of the $K_{soilC,reg}$ parameters that optimize the initial
soil carbon pool sizes.

17    .

**Acknowledgements**
This work was mainly funded by the EU FP7 CARBONES project (contracts FP7-SPACE-
2009-1-242316), with also a small contribution from GEOCARBON project
(ENV.2011.4.1.1-1-283080). This work used eddy covariance data acquired by the
FLUXNET community and in particular by the following networks: AmeriFlux (U.S.
Department of Energy, Biological and Environmental Research, Terrestrial Carbon Program
(DE-FG02-04ER63917 and DE-FG02-04ER63911)), AfriFlux, AsiaFlux, CarboAfrica,
CarboEuropeIP, CarboItaly, CarboMont, ChinaFlux, Fluxnet-Canada (supported by CFCAS,
NSERC, BIOCAP, Environment Canada, and NRCan), GreenGrass, KoFlux, LBA, NECC,
OzFlux, TCOS-Siberia, USCCC. We acknowledge the financial support to the eddy
covariance data harmonization provided by CarboEuropeIP, FAO-GTOS-TCO, iLEAPS, Max
Planck Institute for Biogeochemistry, National Science Foundation, University of Tuscia,
Université Laval and Environment Canada and US Department of Energy and the database
development and technical support from Berkeley Water Center, Lawrence Berkeley National
Laboratory, Microsoft Research eScience, Oak Ridge National Laboratory, University of



California-Berkeley, University of Virginia. P. C. acknowledges support from the European
Research Council through Synergy grant ERC-2013-SyG-610028 "IMBALANCE-P ». The
MODIS MOD09CMG collection 5 surface reflectance data are freely available to download
from the Land Processes Distributed Active Archive Center (LP DAAC) data portal
(https://lpdaac.usgs.gov). The authors wish to thank M. Jung for providing access to the GPP
MTE data, which were downloaded from the GEOCARBON data portal (https://www.bgc-
jena.mpg.de/geodb/projects/Data.php). The authors are also grateful to computing support and
resources provided at LSCE.



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

Memorandum 87802. NASA Goddard Institute for Space Studies, New York, U.S.A.



1 **Tables**

2 Table 1. Parameters description, generality (PFT dependent, global, specific to FLUXNET

3 sites or for a set of regions) and data stream(s) that were used to constrain them.

| Parameter | Description | Dependent | Constraint |
|---|---|---|---|
| $V_{cmax}$ | Maximum carboxylation rate ($\mu$mol·m$^{-2}$·s$^{-1}$) | PFT | Flux, $CO_2$ |
| $G_{s,slope}$ | Ball-Berry slope | PFT | Flux, $CO_2$ |
| $c_{T,opt}$ | Optimal photosynthesis temperature (°C) | PFT | Flux, $CO_2$ |
| $SLA$ | Specific leaf area (m$^2$·g$^{-1}$) | PFT | Flux, $CO_2$ |
| $K_{LAI,happy}$ | LAI threshold to stop using carbohydrate reserves | PFT | Sat, Flux, $CO_2$ |
| $K_{pheno,crit}$ | Multiplicative parameter of the threshold that determines the start of the growing season | PFT | Sat, Flux, $CO_2$ |
| $L_{age,crit}$ | Average critical age of leaves (days) | PFT | Sat, Flux, $CO_2$ |
| $C_{T,sen}$ | Temperature threshold for senescence (°C) | PFT | Sat, Flux, $CO_2$ |
| $F_{stress,h}$ | Parameter reducing the hydric limitation of photosynthesis | PFT | Flux, $CO_2$ |
| $MR_{offset}$ | Offset of the temperature dependence of maintenance respiration | Global | Flux, $CO_2$ |
| $Q10$ | Temperature dependency of heterotrophic respiration | Global | Flux, $CO_2$ |
| $HR_{Hc}$ | Offset of the soil/litter moisture control function | Global | Flux, $CO_2$ |
| $K_{soilC,site}$ | Multiplicative factor of the initial soil carbon pools | per Site | Flux |
| $K_{soilC,reg}$ | | 36 regions | $CO_2$ |
| $K_{albedo}$ | Multiplicative factor of the vegetation albedo | Global | Flux, $CO_2$ |



Table 2. Prior information for all parameters except initial soil C pool multipliers: prior value,
uncertainty and range of variation for the different plant functional types (Tropical Broadleaf
Evergreen/Raingreen forests (TrBE / TrBR), Temperate Needle leaf / Broadleaf Evergreen
forests (TeNE, TeBE), Temperate Broadleaf Deciduous forest (TeBD), Boreal Needle leaf
Evergreen forests (BoNE), Boreal Broadleaf / Needle leaf Deciduous forests (BoBD / BoND)
and C3 grassland.

| Parameter | Plant functional type | | | | | | | | |
|---|---|---|---|---|---|---|---|---|---|
| | TrBE | TrBR | TeNE | TeBE | TeBD | BoNE | BoBD | BoND | NC3 |
| $V_{cmax}$ | 65 ± 24 [35; 95] | 65 ± 24 [35; 95] | 35 ± 12.8 [19; 51] | 45 ± 16 [25; 65] | 55 ± 20 [30; 80] | 35 ± 12.8 [19; 51] | 45 ± 16 [25; 65] | 35 ± 12.8 [19; 51] | 70 ± 25.6 [38; 102] |
| $G_{s,slope}$ | 6.0 ± 2.4 [6; 12] | 6.0 ± 2.4 [6; 12] | 6.0 ± 2.4 [6; 12] | 6.0 ± 2.4 [6; 12] | 6.0 ± 2.4 [6; 12] | 6.0 ± 2.4 [6; 12] | 6.0 ± 2.4 [6; 12] | 6.0 ± 2.4 [6; 12] | 6.0 ± 2.4 [6; 12] |
| $c_{T,opt}$ | 37 ± 6.4 [29; 45] | 37 ± 6.4 [29; 45] | 25 ± 6.4 [17; 33] | 32 ± 6.4 [24; 40] | 26 ± 6.4 [18; 34] | 25 ± 6.4 [17; 33] | 25 ± 6.4 [17; 33] | 25 ± 6.4 [17; 33] | 27.25 ± 6.4 [19.25; 35.25] |
| $SLA$ | 0.015 ± 0.0092 [0.007; 0.03] | 0.026 ± 0.0148 [0.013; 0.05] | 0.009 ± 0.0064 [0.004; 0.02] | 0.02 ± 0.012 [0.01; 0.04] | 0.026 ± 0.0148 [0.013; 0.05] | 0.009 ± 0.0064 [0.004; 0.02] | 0.026 ± 0.0148 [0.013; 0.05] | 0.009 ± 0.0064 [0.004; 0.02] | 0.026 ± 0.0148 [0.013; 0.05] |
| $K_{LAI,happy}$ | 0.50 ± 0.14 [0.35; 0.70] | 0.50 ± 0.14 [0.35; 0.70] | 0.50 ± 0.14 [0.35; 0.70] | 0.50 ± 0.14 [0.35; 0.70] | 0.50 ± 0.14 [0.35; 0.70] | 0.50 ± 0.14 [0.35; 0.70] | 0.50 ± 0.14 [0.35; 0.70] | 0.50 ± 0.14 [0.35; 0.70] | 0.50 ± 0.14 [0.35; 0.70] |
| $K_{pheno,crit}$ | — | 1.0 ± 0.44 [0.7; 1.8] | — | — | 1.0 ± 0.44 [0.7; 1.8] | — | 1.0 ± 0.44 [0.7; 1.8] | 1.0 ± 0.44 [0.7; 1.8] | 1.0 ± 0.44 [0.7; 1.8] |
| $L_{age,crit}$ | 730 ± 192 [490; 970] | 180 ± 48 [120; 240] | 910 ± 240 [610; 1210] | 730 ± 192 [490; 970] | 180 ± 48 [120; 240] | 910 ± 240 [610; 1210] | 180 ± 48 [120; 240] | 180 ± 48 [120; 240] | 120 ± 60 [30; 180] |
| $C_{T,sen}$ | — | — | — | — | 12 ± 8 [2; 22] | — | 7 ± 8 [−3; 17] | 2 ± 8 [−8; 12] | −1.375 ± 8 [−11.375; 9.375] |
| $F_{stress,h}$ | 6.0 ± 3.2 [2; 10] | 6.0 ± 3.2 [2; 10] | 6.0 ± 3.2 [2; 10] | 6.0 ± 3.2 [2; 10] | 6.0 ± 3.2 [2; 10] | 6.0 ± 3.2 [2; 10] | 6.0 ± 3.2 [2; 10] | 6.0 ± 3.2 [2; 10] | 6.0 ± 3.2 [2; 10] |
| $MR_{offset}$ | 1.0 ± 0.6 [0.5; 2.0] | | | | | | | | |
| $Q10$ | 1.99372 ± 0.8 [1.0; 3.0] | | | | | | | | |
| $HR_{Hc}$ | −0.29 ± 0.24 [−0.59; 0.01] | | | | | | | | |
| $K_{albedo}$ | 1.0 ± 0.16 [0.8; 1.2] | | | | | | | | |



1  **Figures**

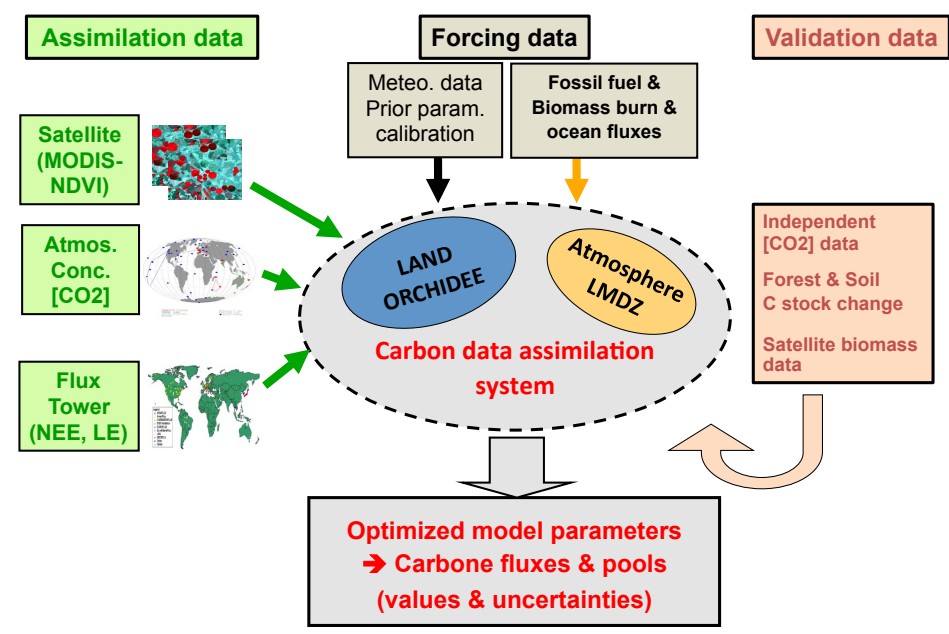

7   Figure 1. Schematic of the ORCHIDEE Carbon Cycle Data Assimilation System
8   (ORCHIDAS).





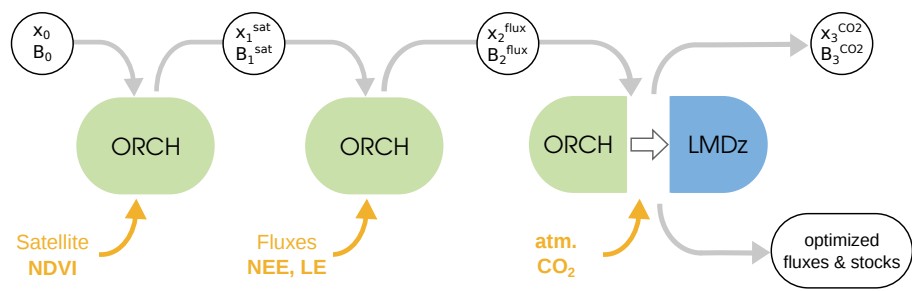

2    Figure 2. Illustration of the step-wise data assimilation approach used for the assimilation of
3    multiple data streams in the ORCHIDEE-CCDAS. The list of parameters for each step is
4    summarized in Table 1.

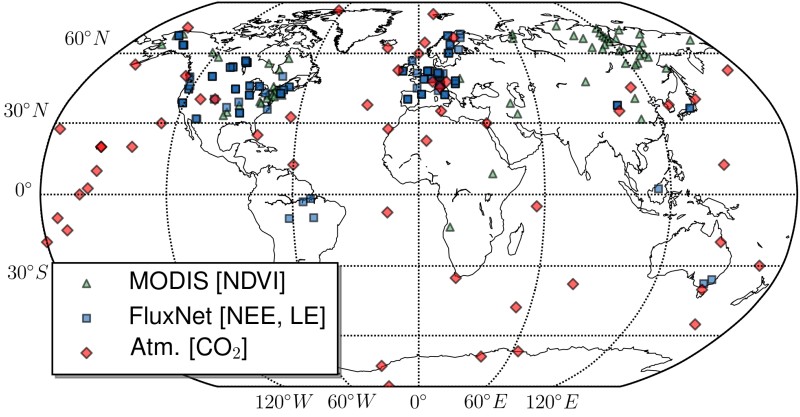

9    Figure 3: Location of the different observations used for each data stream assimilated in the
10   system: MODIS-NDVI measurements, FLUXNET sites with NEE and LE measurements and
11   atmospheric $CO_2$ stations.





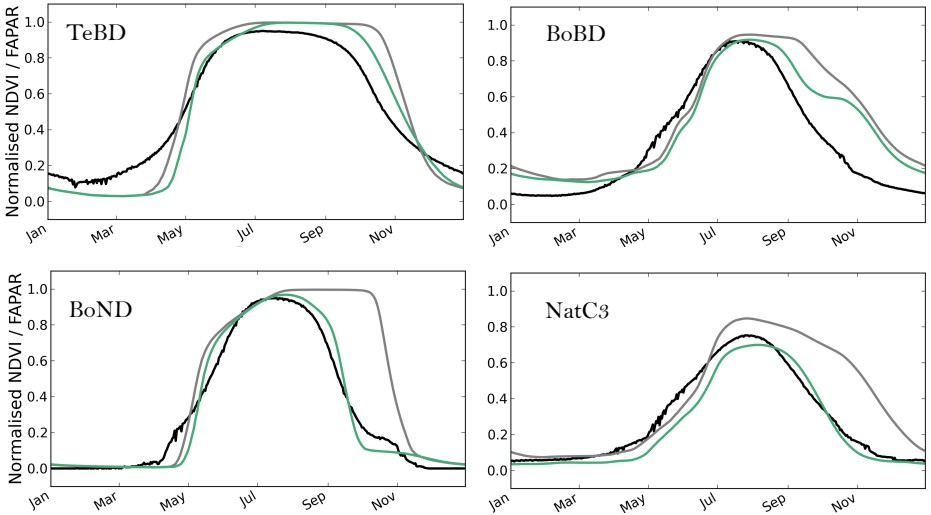

Figure 4. Mean seasonal cycle (2000-2008) of the normalised modelled FAPAR before and
after optimisation, compared to that of the MODIS NDVI data, for the temperate and boreal
deciduous PFTs (TeBD, BoBD, BoND and NatC3). Black = MODIS NDVI data; Grey =
prior simulation (default ORCHIDEE parameters); Green = posterior multi-site optimisation.

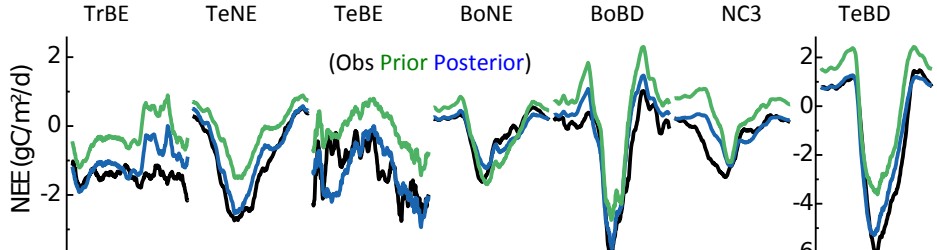

Figure 5: Mean seasonal cycle of the Net Carbon Ecosystem Exchange (NEE) for the
different plant functional type optimize in Step 2 of the assimilation. The mean across all sites
for a given PFT is provided for the observations (black), the prior ORCHIDEE (grey), the
posterior of step 1 (green) and the posterior of step 2 (blue).

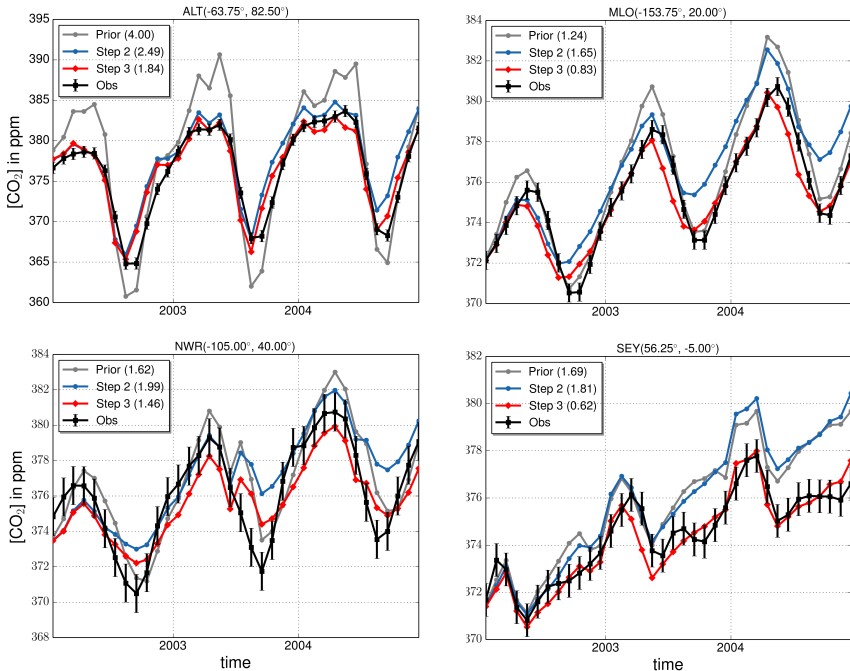

Figure 6: Monthly mean atmospheric $CO_2$ concentrations after step 3 of the optimization, for
several stations over the period 2002-2004 of the optimization. The observations (black), the
prior model (grey) and the posterior model after step 2 (blue) and step 3 (red) are displayed.
Numbers in parenthesis correspond to RMSEs.

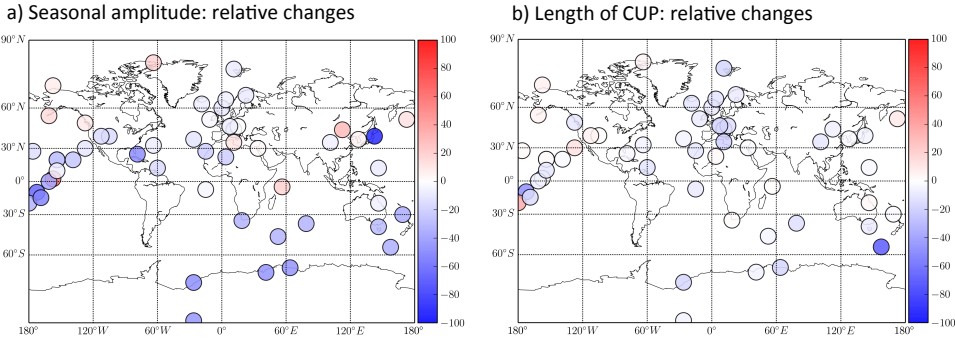





Figure 7: Changes in the mean seasonal cycle of the atmospheric $CO_2$ concentrations after
step 3 of the optimization at all atmospheric stations. Left: Relative changes (in percentage)
between the prior and posterior absolute model-data differences for the amplitude of the
seasonal cycle. Right: Same metric but for the length of the Carbon Uptake Period (CUP),
measured as the sum of the days when the de-trended concentrations are negative (see text).

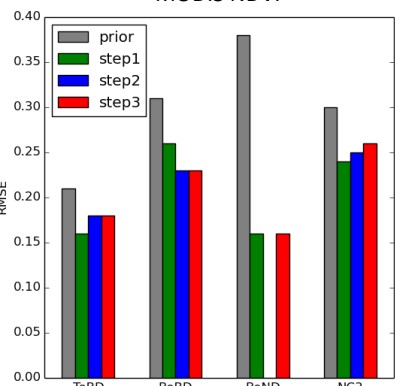
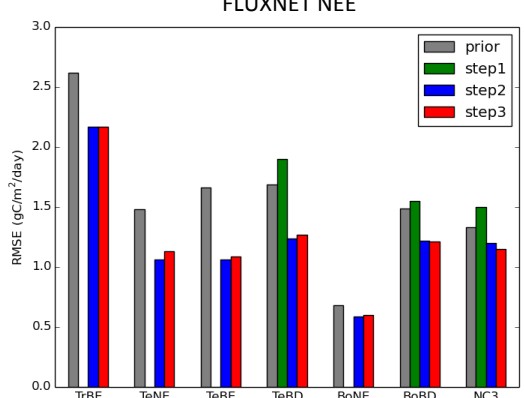

Figure 8: RMSE between model outputs and observations for two types of observations:
MODIS-NDVI on the left and FluxNet-NEE on the right, for different Plant Functional Types
(PFT: TrBE, TeNE, TeBE, TeBD, BoBD, BoND, NC3) and for the prior model simulation
and the posterior of each step of the assimilation framework. Missing bars correspond to the
fact that no data were available to constrain a given PFT.

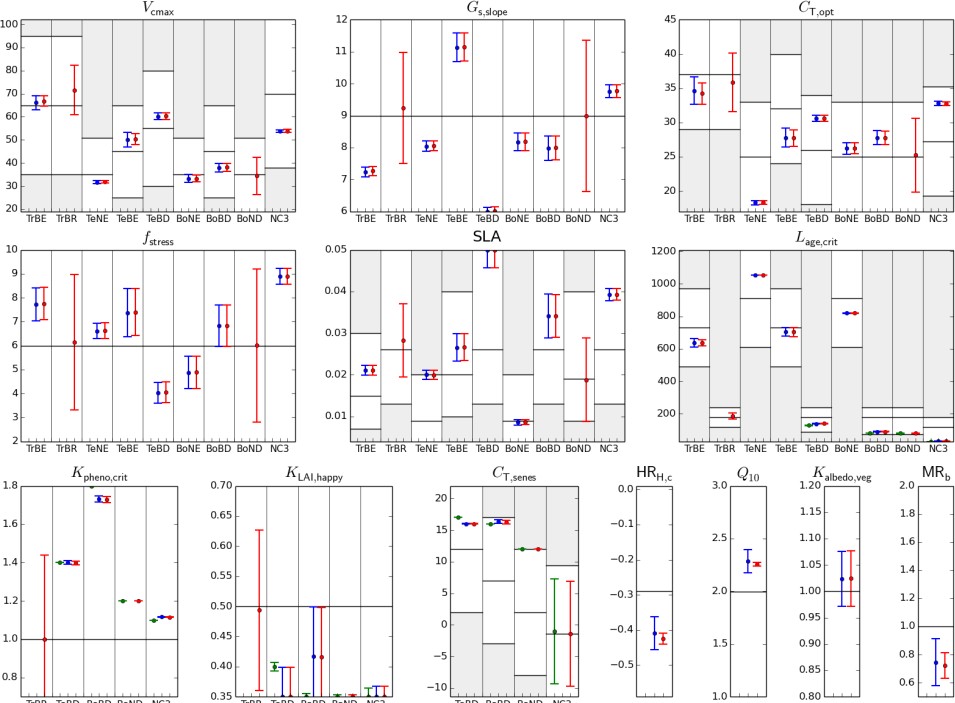

Figure 9: Prior and posterior parameter values and uncertainties for a set of optimized parameters (9 PFT dependent and 4 non-PFT dependent). The prior value corresponds to the horizontal black line and the physical allowed range of variation to the "y" range (i.e. the white zone). For PFT-dependent parameters, there are 9 sub-plots corresponding to PFTs that were optimized (except for $K_{pheno\_crit}$ with only 5 PFTs). For each parameter, there are 3 estimated values for the three successive steps: step1: assimilation of MODIS-NDVI data (green symbol); step2: adding FLUXNET data (blue symbol); step3: adding atmospheric $CO_2$ data (red symbol). The parameter values are depicted with the symbols and the estimated uncertainties with the vertical line (± sigma).





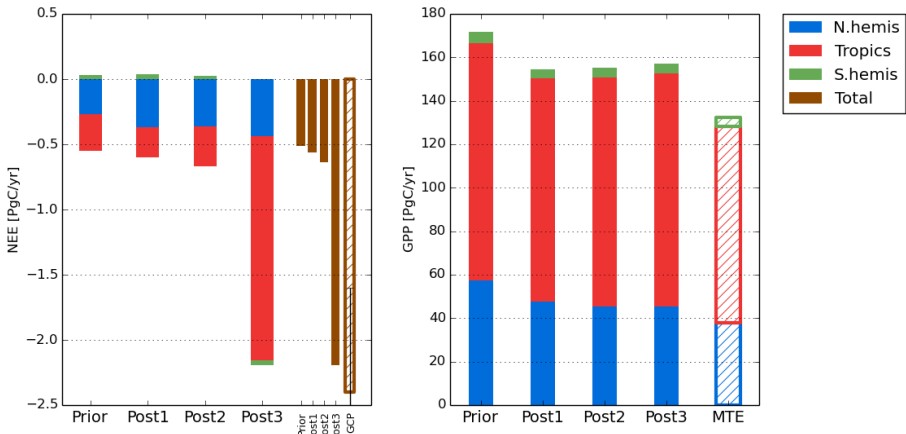

Figure 10: Left: Net Ecosystem Exchange (NEE) for three regions (North of 35°N, Tropics,
South of 35°S) for the prior model, and after each step of the optimizations (mean over 2002-
2004). The total NEE is indicated with the vertical brown bar and compared to the Global
Carbon Project (GCP) estimate for the same period (Le Quéré et al. 2015). Right: same but
for Gross Primary Production where the data driven estimate (MTE product using FluxNet
data; Jung et al., 2009) is provided for comparison.



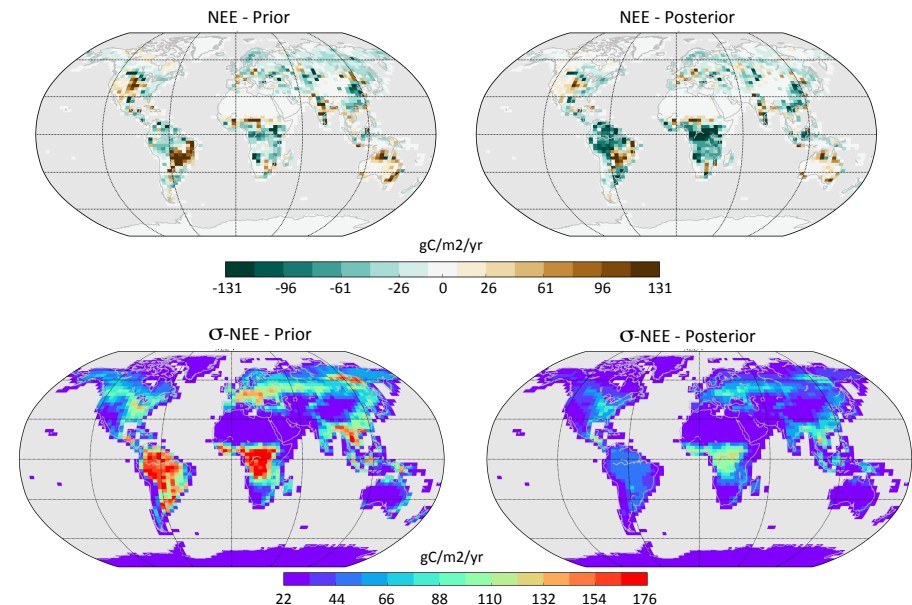

Figure 11: Simulated annual net carbon exchange (NEE) for the land ecosystems prior to any
optimization (left column) and after step 3 of the optimization process (right column). Upper
figures correspond to the mean NEE (in $gC.m^{-2}.y^{-1}$) over the assimilation period (2001-2003)
and lower figures to the associated monthly flux uncertainties (averaged over the whole
period and expressed in $gC.m^{-2}.y^{-1}$) due to the parameter uncertainties (see text).





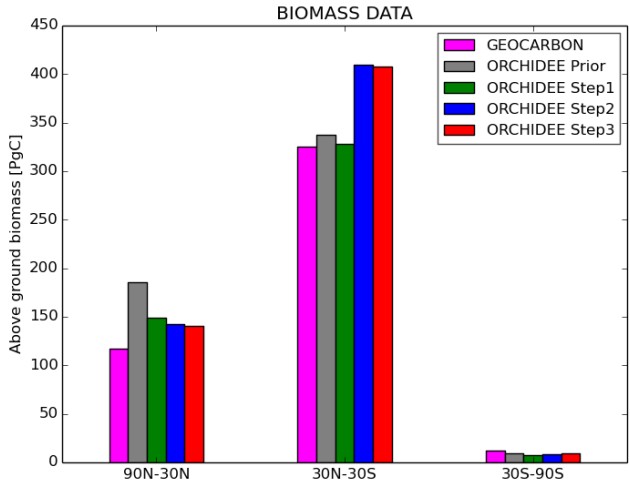

Figure 12: Above ground forest biomass data for the prior ORCHIDEE model and after step
1, step 2 and step 3 of the optimization process. Estimates from satellite observations (Santoro
et al., 2015) and referred as "GEOCARBON" (following the EU-GEOCARBON project) are
provided for comparison.





# 1 Appendix figures

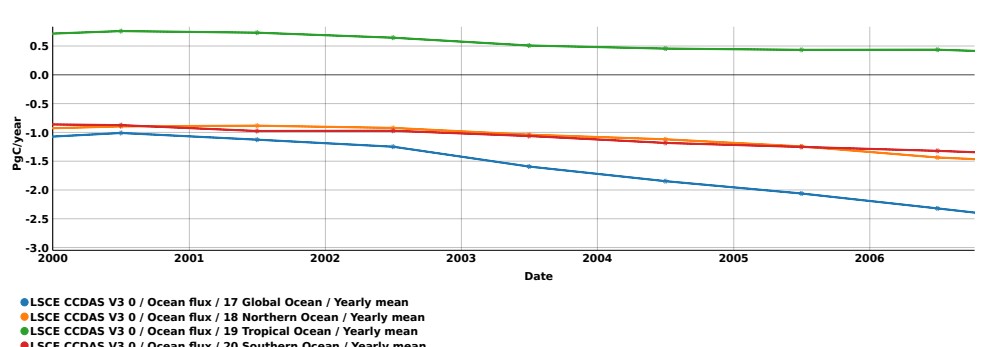

Figure A1: $CO_2$ air-sea fluxes including the natural ocean out-gazing, used as input to the ORCHIDEE-CCDAS and estimated from a neural network approach using observed $pCO_2$ data (see main text, section 2.5.1). The Northern, Tropical and Southern ocean contributions to the global ocean flux (blue curve) are also provided.

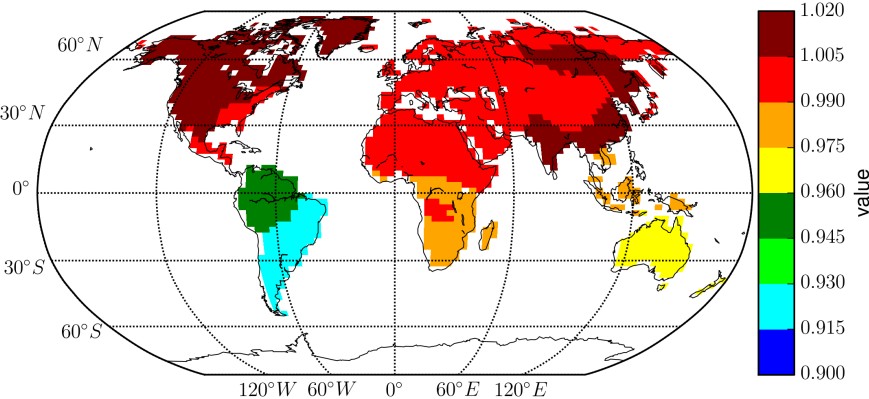

Figure A2: Map of the posterior values of the coefficient scaling the initial carbon pool sizes per regions.

