# Peer review of "A new step-wise Carbon Cycle Data Assimilation System using multiple data streams to constrain the simulated land surface carbon cycle"

_Geoscientific Model Development, 2016_

## Referee Comment (RC1) · Anonymous Referee #1 · 23 Feb 2016

Peylin and colleagues describe a Carbon Cycle Data Assimilation System based on the land-surface model ORCHIDEE optimized against NDVI data, eddy covariance $CO_2$ flux data, and atmospheric $CO_2$ data. For practical reasons, these three data streams are used successively in three steps. The paper describes the system, assesses its performance (especially the self-consistency across the three steps), and some features of the resulting carbon cycle fluxes and stocks. The authors conclude that the ORCHIDEE land-surface model is now structurally adquate enough to bridge the information from the three data streams, though they also highlight further steps that need to be taken to represent the global carbon cycle more accurately.

The study represents an interesting and relevant development in the understanding of

the carbon cycle consistent with available data. There are open issues (for example the short assimilation period precluding various processes to be constrained and assessed) but these are clearly acknowledged in the paper. I find the presentation clear and convincing. In my opinion, the work should be published in Geoscientific Model Development.

Minor comments:

p 5 l 6: The associativity is true for linear systems, but is it really also true for non-linear systems? (I nevertheless agree to the arguments given in favor of the step-wise approach.)

p 8 l 13: What is the uncertainty due to incomplete sampling of the diurnal cycle?

p 14 l 25-26: I was wondering whether the presence of step functions, creating discontinuities, still allows a well-defined solution of the minimization?

p 17 l 25-30: Add references for "University of Stuttgart" and "ENSTO-E". Explain abbreviation "IER".

p 20 l 14: What does "conditions" mean here?

p 21 l 4: Clarify whether this is the prior before step 1 or before step 3?

p 22 l 21: Can you give just a brief summary of the reasons here?

p 25 l 24: These are clearly not the numbers shown in Fig 10 right.

p 27 l 6: Can you explain (here or earlier) why you used individual grid points rather than the whole grid?

p 29 l 10: I think you should also mention the errors in the prescribed fossil fuel and ocean fluxes.

Fig 5: I don't see any grey lines.

Typos and suggested formulations:

p 2 l 7: "uncertainties of simulated carbon fluxes and stocks"

p 3 l 15: "Carbon cycle componenets"

p 7 l 13: "propagate"

p 8 l 7: missing "-"?

p 10 l 1: "the third step"

p 12 l 5: suprious "the"

p 17 eq 7: The first ")" seems to belong to the index. There seems to be a "," missing before "LAT".

p 17 l 19 "outgassing"

p 18 l 29 Why "Fig 8"?

p 22 l 17: "from" rather than "between"?

Fig 5 caption: "optimized"

Fig 7 caption: prior before step 1 or before step 3?

[Figure]

---

## Referee Comment (RC2) · J.-F. Exbrayat (Referee) · 3 Mar 2016

Peylin et al. contribute an interesting study on the effect of using a stepwise optimisation rather than merging all data streams in a single cost function. This is a procedure that we have used in e.g. catchment scale water quality modelling where one first calibrates the water cycle before calibrating parameters relevant to nutrient diffusion (e.g. Exbrayat et al., 2011). However, this approach has not been investigated in details in the frame of the (global) carbon cycle. Therefore, I agree with reviewer #1 that this paper is highly relevant to the community. I particularly like the several steps used by the authors to reconcile site-scale calibration with global atmospheric concentrations. I have some very minor comments on the paper that should be straightforward

to address:

p.5 l.1: another alternative to stepwise optimisation and simultaneous procedure is a simultaneous, multi-objective approach based on the Pareto ranking of several cost functions to account for trade-offs (e.g. Yapo et al., 1998). Would it be realistic to use such an approach in this system to avoid the increase in RMSE against MODIS NDVI from step1 to step2 (fig. 8, TeBD)?

p.5 l.12-13: Using a restricted number of parameters is a valid point but it needs to be mentioned here that one must proceed to some sort of sensitivity analysis to accurately select these parameters.

p.8 l.1: why not using only days with data?

p.9 l.26: please mention the resolution of the model here

p. 10 l.1: is it robust to assume that carbon pools are at equilibrium in 1990? Could this system use a prior from soil and biomass maps instead (like e.g. Bloom et al., 2016)?

p. 18 l.6: How is fire simulated during spin-up?

p.26 l.29: perhaps "ecosystem data streams" is more correct (LE is not carbon sensu stricto)

p. 29 l.17: see also Bloom et al. (2016)

Figure 1: lower box, "carbon"

Figure 5: please add scale on x-axis

References

Bloom, A. A., Exbrayat, J.-F., van der Velde, I. R., Feng, L. and Williams, M.: The decadal state of the terrestrial carbon cycle: Global retrievals of terrestrial carbon allocation, pools, and residence times., Proc. Natl. Acad. Sci. U. S. A., 113(5), 1285–1290, doi:10.1073/pnas.1515160113, 2016.

Exbrayat, J.-F., Viney, N. R., Frede, H.-G. and Breuer, L.: Probabilistic multi-model ensemble predictions of nitrogen concentrations in river systems, Geophys. Res. Lett., 38, L12401, doi:10.1029/2011GL047522, 2011.

Yapo, P.O., Gupta, H.V. and Sorooshian, S., 1998. Multi-objective global optimization for hydrologic models, J. Hydrol., 204, 83-97.

---

## Referee Comment (RC3) · Anonymous Referee #3 · 4 Apr 2016

General comments

The manuscript reports development and application of a data assimilation system which is used to produce a version of ORCHIDEE model optimized to reproduce NDVI, net ecosystem exchange and latent heat flux at land validation/flux tower sites and $CO_2$ seasonality at background $CO_2$ monitoring sites. The stepwise optimization approach is proposed as a simplified alternative to optimizing model to fit NDVI, flux tower data and atmospheric $CO_2$ data simultaneously. Despite splitting the process in several stages authors succeeded to find a set of parameters allowing the model to fit all types of constraints.

The manuscript is well written, and presents an original and valuable contribution. It

can be published after minor revision, hopefully addressing the comments listed below.

General comment 1. In the optimization framework adopted by authors, model parameters optimised at the set of flux tower sites are later extrapolated to whole land surface using available spatial data on vegetation type, weather and soil type information as drivers. The flux tower site optimization is made by combining several sites within same vegetation type in one group, and average flux seasonality is shown to be improved by the optimization. The variability of the fluxes due to soil quality and slope/drainage within same vegetation type is not directly captured by this approach, while some studies (Ise an Sato, 2008) suggest there is a way to address site level differences in productivity potential (edaphic variability) based on remote sensing data. It would be relevant to mention this factor in discussing reasons for remaining spread in the degree of success that can be achieved using one set of model parameters for optimizing fluxes at several sites of same vegetation type.

Detailed comments

Page 01 – Line 03. In addition to "incorrect model parameter values" one should mention uncertainty in spatial distribution of the parameters coming from the maps of soil properties, topographic features, vegetation types.

Page 04 – Line 05. Randerson et al (1996) paper can be mentioned among influential studies that use air concentration as constraint

Page 04 – Line 26 To extend a list studies using multiple input streams and C stock data in assimilation (Saito et al 2014) can be added.

Page 20 Line 13. In many transport models it appears difficult to match CO2 seasonal cycles in PBL and free troposphere at the same time, which can be attributed to simulated PBL height biases and biases in other processes. The problem can lead to finding a set of model parameters that are optimized well for LMDz model with its PBL height and PBL ventilation rate, but not performing that well when model is different.

[Figure]

It would be useful to add figure showing match with free tropospheric data for model validation. Aircraft data and TCCON data can be used for validation, especially high latitude sites know for high seasonal amplitude such as Poker Flats Alaska, or TCCON at Sodankula (Lindqvist et al., 2015). A useful check would include use of vertically integrated profile data at airborne observation sites (Nakatsuka and Maksyutov, 2009), as it is more stable against the PBL height biases.

Technical corrections

Fig 1. Correct spell: "Carbone fluxes" to "Carbon fluxes"

References

Ise, T., and H. Sato : Representing subgrid-scale edaphic heterogeneity in a large-scale ecosystem model: A case study in the circumpolar boreal regions, Geophys. Res. Lett., 35, L20407, doi:10.1029/2008GL035701, 2008.

Lindqvist, H. and coauthors: Does GOSAT capture the true seasonal cycle of carbon dioxide?, Atmos. Chem. Phys., 15, 13023-13040, doi:10.5194/acp-15-13023-2015, 2015.

Nakatsuka, Y. and Maksyutov, S.: Optimization of the seasonal cycles of simulated $CO_2$ flux by fitting simulated atmospheric $CO_2$ to observed vertical profiles, Biogeosciences, 6, 2733-2741, doi:10.5194/bg-6-2733-2009, 2009.

Randerson, J. T. and coauthors: Substrate limitations for heterotrophs: Implications for models that estimate the seasonal cycle of atmospheric $CO_2$, Global Biogeochem. Cycles, 10(4), 585–602, doi:10.1029/96GB01981, 1996

Saito, M., and coauthors: Optimization of a prognostic biosphere model for terrestrial biomass and atmospheric $CO_2$ variability, Geosci. Model Dev., 7, 1829-1840, doi:10.5194/gmd-7-1829-2014, 2014.

---

## Referee Comment (RC4) · Anonymous Referee #4 · 27 Apr 2016

**Comment on P. Peylin et al., "A new step-wise Carbon Cycle Data Assimilation System using multiple data streams to constrain the simulated land surface carbon cycle" by Anonymous Referee 4**

**General Comments**

The manuscript presents a sequence of parameter estimation exercises for the ORCHIDEE Land Surface Model using a CCDAS data assimilation framework. Firstly, NDVI data are assimilated at point scale. Secondly, FLUXNET data are assimilated at point scale. Thirdly, atmospheric CO2 data are assimilated at global scale.

The presentation of the material is excellent, despite some minor inconsistencies.

The novelty of the material is limited.

What the authors present as a step-wise system, are in fact three systems that are operated in a sequence. The interface between these systems is minimal: It consist of selected parameters with error bars but excluding the error covariance that are passed in one direction.

The step-wise approach is not new. It is desribed, for example, by Rayner et al. (2005): They assimilate NDVI in the first step and atmospheric CO2 in the second step.

The system for assimilation of NDVI is described in more detail elsewhere (MacBean et al., 2015).

The system for assimilation of FLUXNET data is described in more detail elsewhere (Kuppel et al., 2012, 2014).

What is left is the system for assimilation of a single data stream, i.e. the atmospheric CO2 data from 2002 to 2004. The description of the assimilation method is provided elsewhere (see above references). The ORCHIDEE LSM, the LMDz CTM and the use of influence functions was also described elsewhere (see references in section 2.3.2). The assimilation of atmospheric CO2 using a combination of an LSM and a CTM and prescribed emissions from other components of the carbon cycle is not new either. It was presented by Rayner et al. (2005) and applied for a time span of two decades.

In summary the manuscript is not suitable for GMD because it fails to present "substantial new concepts, ideas, or methods".

The scientific approach of passing reduced information on the parameters from one assimilation system to the next is questionable. The reviewer agrees with the author's statement: "It is important to note that this is an implementation question. Tarantola (2005) recasts the fundamentals of the approach as the conjunction or multiplication of probability densities. This multiplication is associative so it makes no difference whether it is performed in one step or several." However, an implementation of such a

step-wise procedure needs to propagate the full Probably Density Function from one step to the next. In the Gaussian framework selected here this requires to propagate the full error covariance matrix and not just the error bars (see comment above). Error correlations are to be expected (see, e.g. Raoult et al., 2016). The change of the parameter space from one step to the next adds a further weakness as well as the dependence of $H_\infty$ on the last iteration of each step. The degradation of the results in the back-compatibility test is no surprise. Another test that has not been performed here would be to operate the sequence of assimilation systems in the reverse order and compare the final parameters and validation results. The computing effort is the same as for the order presented here.

The assimilation of a statistical index, i.e. NDVI, is somewhat beyond state of the art, as assimilation of the related physical variable, FAPAR, has been demonstrated for multiple LSMs (Knorr et al., 2010, Schurmann et al, 2016). The required physical model of FAPAR is available in ORCHIDEE (Naudts et al., 2015).

**Specific Comments**

p11: 184 parameters is misleading, as none of the three systems estimates that many parameters

Why are $K_{soilC}$ parameters differentiated per region?

p23: Why are the FLUXNET assimilations performed per site and not simultaneously? How is the error of the parameter averaged over PFTs calculated.

Eq.(1) in the manuscript does not correspond with Eq. (1) in Tarantola (1987).

p21: After assimilation of atmospheric CO2 it is no surprise that the trend is close to observations.

p24: Fluxes are calculated from 2000 to 2009. Why are concentrations in Figure 6 not shown over the same time span?

[Figure]

p41: 36 regions while in text it is 30.

**Technical Corrections**

p e l 16: "remains"change to "remain"

p 13 l 13: "we did not propagated" change to "we did not propagate"

p 9 l 12: "et al., (1980)" change to "et al. (1980)"

**References**

References from manuscript not duplicated.

Knorr, W., T. Kaminski, M. Scholze, N. Gobron, B. Pinty, R. Giering, and P.-P. Mathieu, Carbon cycle data assimilation with a generic phenology model, J. Geophys. Res., 115, G04017, doi:10.1029/2009JG001119, 2010.

Raoult, N. M., Jupp, T. E., Cox, P. M., and Luke, C. M.: Land surface parameter optimisation through data assimilation: the adJULES system, Geosci. Model Dev. Discuss., doi:10.5194/gmd-2015-281, in review, 2016.

Schurmann, G. J., Kaminski, T., Kostler, C., Carvalhais, N., Voßbeck, M., Kattge, J., Giering, R., Rodenbeck, C., Heimann, M., and Zaehle, S.: Constraining a land surface model with multiple observations by application of the MPI-Carbon Cycle Data Assimilation System, Geosci. Model Dev. Discuss., doi:10.5194/gmd-2015-263, in review, 2016.
* * *

---

## Author Comment (AC1) · 12 Jun 2016

Peylin and colleagues describe a Carbon Cycle Data Assimilation System based on the land-surface model ORCHIDEE optimized against NDVI data, eddy covariance $CO_2$ flux data, and atmospheric $CO_2$ data. For practical reasons, these three data streams are used successively in three steps. The paper describes the system, assesses its performance (especially the self-consistency across the three steps), and some features of the resulting carbon cycle fluxes and stocks. The authors conclude that the ORCHIDEE land-surface model is now structurally adequate enough to bridge the information from the three data streams, though they also highlight further steps that need to be taken to represent the global carbon cycle more accurately.

[Figure]

The study represents an interesting and relevant development in the understanding of the carbon cycle consistent with available data. There are open issues (for example the short assimilation period precluding various processes to be constrained and assessed) but these are clearly acknowledged in the paper. I find the presentation clear and convincing. In my opinion, the work should be published in Geoscientific Model Development.

We thank the reviewer for his positive appreciation of the manuscript.

Minor comments:

p 5 l 6: The associativity is true for linear systems, but is it really also true for nonlinear systems? (I nevertheless agree to the arguments given in favor of the step-wise approach.)

The associativity argument, detailed in Tarantola (2005), relies on the combination of probability density functions (PDFs) through the Bayes theorem without any assumption on the linearity or non-linearity of the system. It is a general property associated to the combination of probability distribution (i.e., the description of the probability of an event, based on conditions that might be related to that event). So the non-linearity does not invalidate the step-wise approach; it only highly complicates the computation of the full PDFs at each step and thus their propagation. The necessary simplifications that are made in this case lead to the complication investigated in the paper. In the case of a linear system, we could have easily calculated and propagated the full PDFs. We have completed one sentence in the introduction to reinforce that the associativity does not depend on the linearity of the system: "This multiplication is associative so it makes no differences whether it is performed in one step or several (and whether the system is linear or not)."

p 8 l 13: What is the uncertainty due to incomplete sampling of the diurnal cycle?

We have not estimated the uncertainty due to a possible 20% gap. However, in order

to compute the daily mean values, we have used gap-filled data so that the uncertainty only arises due to errors in the gap-filling procedure. Such uncertainty, usually less than 20 % (Lasslop et al. 2008, would thus only affect 20% at maximum of the diurnal cycle; it would thus have an overall negligible impact compared to the model uncertainty that need to be also accounted for.

p 14 l 25-26: I was wondering whether the presence of step functions, creating discontinuities, still allows a well-defined solution of the minimization?

We agree that step functions may complicate the minimization problem, creating potentially non-smooth cost function, i.e. with a "singular" point where the derivative is not defined and not continuous. However the use of finite differences to compute the gradient for these parameters allows defining a "mean derivative" at any point (mean as for non local). As a result the iterative approach (BFGS algorithm) used to search for the minimum may oscillate if we end up for a given parameter in the vicinity of the discontinuity, but it will not diverge. However such situation is very unlikely to occur and we have checked that no obvious oscillation for each parameter was occurring. We completed one sentence in the text to precise this point: "A finite difference approach was used for these parameters in order to define a mean derivative at any point".

p 17 l 25-30: Add references for "University of Stuttgart" and "ENSTO-E". Explain abbreviation "IER".

We now explain the "IER" abbreviation and we provide two web references for both "University of Stuttgart" and "ENSTO-E".

p 20 l 14: What does "conditions" mean here?

We have change the text to precise that "conditions" was referring to station sampling air "representative of different geographical regions of influence".

p 21 l 4: Clarify whether this is the prior before step 1 or before step 3?

It is the prior of step 3 and we have now clarified the text.

p 22 l 21: Can you give just a brief summary of the reasons here?

We have improved the text to provide a brief summary of these reasons: • The NDVI or fAPAR mainly constrain the timing of the ecosystem GPP and only to a small extend the amplitude of the GPP, given that the satellite signal is likely to saturate during the peak of the growing season. • The NDVI or fAPAR do not constraint at all the ecosystem respiration, which is an important component of the NEE at least at seasonal to annual scales.

p 25 l 24: These are clearly not the numbers shown in Fig 10 right.

We apologize for such mistake as the numbers correspond indeed from a previous experiment. We have corrected the text with the exact numbers from figure 10b. Note that the exact numbers from figure 10b do not change the overall message and the rather small reduction of the GPP from the prior to the posterior of step 3.

p 27 l 6: Can you explain (here or earlier) why you used individual grid points rather than the whole grid?

The main reasons for choosing only a set of individual grid points are twofold: • First, we proposed to used only the model grid point that are covered by a vegetation fraction greater than 60% for a given PFT, avoiding the use of grid points with a large mix of PFTs so that the optimization of the phenological parameters for each PFT is more straightforward. This is explained in section 2.4.1 (step 1). • We then limited the selected set of points to 15 in order to significantly reduce the computing time of the step 1 optimization. Doing so further allows evaluating the optimized model at all pixels that were not used for the optimization. We added a sentence in section 2.4.1 (step 1 paragraph) to better explain our choice.

p 29 l 10: I think you should also mention the errors in the prescribed fossil fuel and ocean fluxes.

We added at the end of the paragraph a sentence to mention the importance of the

errors associated to the prescribed fossil fuel and ocean fluxes.

Fig 5: I don't see any grey lines.

We have corrected the caption as we do not included the "prior of step 1" in this version of the figure in order not to overload the figure with too many curves.

Typos and suggested formulations:

p 2 l 7: "uncertainties of simulated carbon fluxes and stocks"

Corrected

p 3 l 15: "Carbon cycle componenets"

Corrected

p 7 l 13: "propagate"

Corrected

p 8 l 7: missing "-"?

Corrected

p 10 l 1: "the third step"

Corrected

p 12 l 5: suprious "the"

Corrected

p 17 eq 7: The first ")" seems to belong to the index. There seems to be a "," missing before "LAT".

Corrected

p 17 l 19 "outgassing"

Corrected

p 18 l 29 Why "Fig 8"?

We wanted to refer to the figure displaying the parameter values before and after the optimization. This figure is Fig. 9 and not Fig. 8. We have thus corrected.

p 22 l 17: "from" rather than "between"?

Corrected

Fig 5 caption: "optimized"

Corrected

Fig 7 caption: prior before step 1 or before step 3?

We have précised that it is the prior of step 3.
* * *

---

## Author Comment (AC4) · 14 Jun 2016

General comments: The manuscript presents a sequence of parameter estimation exercises for the ORCHIDEE Land Surface Model using a CCDAS data assimilation framework. Firstly, NDVI data are assimilated at point scale. Secondly, FLUXNET data are assimilated at point scale. Thirdly, atmospheric CO2 data are assimilated at global scale. The presentation of the material is excellent, despite some minor inconsistencies.

We thank the reviewer for having commented our manuscript. We explain below why we disagree to some of his major comments.

The novelty of the material is limited. What the authors present as a step-wise system, are in fact three systems that are operated in a sequence. The interface between these systems is minimal: It consist of selected parameters with error bars but excluding the error covariance that are passed in one direction. The step-wise approach is not new. It is desribed, for example, by Rayner et al. (2005): They assimilate NDVI in the first step and atmospheric CO2 in the second step. The system for assimilation of NDVI is described in more detail elsewhere (MacBean et al., 2015). The system for assimilation of FLUXNET data is described in more detail elsewhere (Kuppel et al., 2012, 2014). What is left is the system for assimilation of a single data stream, i.e. the atmospheric CO2 data from 2002 to 2004. The description of the assimilation method is provided elsewhere (see above references). The ORCHIDEE LSM, the LMDz CTM and the use of influence functions was also described elsewhere (see references in section 2.3.2). The assimilation of atmospheric CO2 using a combination of an LSM and a CTM and prescribed emissions from other components of the carbon cycle is not new either. It was presented by Rayner et al. (2005) and applied for a time span of two decades. In summary the manuscript is not suitable for GMD because it fails to present "substantial new concepts, ideas, or methods".

We disagree with the main criticism that our study does not provide new ideas or methods. In order to explain this further we need to lay out the evolving state of carbon cycle data assimilation.

Systems that apply the well-established methods of data assimilation to models of the carbon cycle at various scales have been around for nearly two decades. Wang et al. (2001) and Kaminsky et al (2002) antecede Rayner et al. (2005). The problem pointed out by Rayner et al. (2010) is that information was not transferable between either different sites or different datastreams. Rayner et al. (2005) for example, did not expose any of the phenological parameters of the assimilation from Knorr et al. (2001) in their assimilation so there could be no test of consistency. Rayner et al. (2010) pointed out that evolution of both models and methods was necessary for comprehensive assimi-
lation. The current paper represents an important step in that evolution though by no means the final one.

 c The paper describes for the first time (to our knowledge) a system that is able to assimilate three major carbon cycle data streams (vegetation activity from satellite, FluxNet data and atmospheric CO2) in a process-based land surface model used as the land component of an Earth System Model (ESM). No such system has been described so far, although this is a major challenge given the differences obtained for the carbon cycle in the last collection of model used for CMIP5 exercise (last IPCC report).

 c The reviewer slightly overstates the achievement of Rayner et al. (2005). Although it did use soil moisture and radiation fields from an earlier assimilation from a related model (a simpler version) this was irrelevant to the narrative of that paper. The fields could just as easily have come from a direct satellite product. There was little comment made on the consistency between the two assimilations and no parameters passed between them. Since then, numerous global scale carbon cycle data assimilation studies have been published (above 10) all of them contributing additional aspects to these systems. However in those 10 years a third and crucial data stream, namely the widely used FLUXNET network of net CO2 and latent heat flux observations, has not yet been included with FAPAR and CO2 data in a global scale assimilation. All other studies to our knowledge use FAPAR data and CO2 or FLUXNET data, but none have used all three.

 c The focus of the paper lies in the combination of these three data streams while the individual papers MacBean et al. 20015 and Kuppel et al. 2012, 2014 focus on the impact of each individual data stream. The major result of the paper is that for the first time "a state of the art global land surface model part of an ESM is able to capture with a reasonable accuracy the information content of three major data streams following a data assimilation procedure". This is no insignificant feat as it opens new perspectives to reduce the spread of the land carbon sink simulated by the CMIP5 suite of models

and thus to reduce possibly the uncertainty in long-term climate predictions.

• We acknowledge that the chosen approach may not be the optimal one in a statistical sense with only the propagation of the error variance of the optimized parameters (see more detail in the response to the next comment).

• We and the reviewer agree that this is, fortunately, a burgeoning field of activity. Raoult et al (2016) and Sherman et al. (2016) are contemporarily studies, that rely on land surface models of an ESM. They present different strengths and weaknesses from this paper. However we are confident that the paper does represent an advance in available methods. Raoult, et al. (2016) only uses FluxNet observations to optimize the parameters of the JULES model, while Schürmann et al. (in revision) only assimilate two data streams (fAPAR and CO2) in JSBACH model (at coarse resolution, $10° \times 10°$). Note finally that the level of complexity of the ecosystem model is part of the problem: achieving an optimization with a simpler model does not guaranty that the framework would work with a more complex one.

Overall, the paper relies on old data assimilation concepts (published way before Rayner et al. 2005) but provides a new implementation (3 data streams with a state of the art component of an ESM) and opens the road for improved carbon – climate coupled simulations and improved climate predictions.

We acknowledge that we have not done enough to highlight the new features of this study, and thus we have emphasized these points in the "introduction" and "discussion and conclusion" sections.

The scientific approach of passing reduced information on the parameters from one assimilation system to the next is questionable. The reviewer agrees with the author's statement: "It is important to note that this is an implementation question. Tarantola (2005) recasts the fundamentals of the approach as the conjunction or multiplication of probability densities. This multiplication is associative so it makes no difference whether it is performed in one step or several." However, an implementation of such a

step-wise procedure needs to propagate the full Probably Density Function from one step to the next. In the Gaussian framework selected here this requires to propagate the full error covariance matrix and not just the error bars (see comment above). Error correlations are to be expected (see, e.g. Raoult et al., 2016). The change of the parameter space from one step to the next adds a further weakness as well as the dependence of H1 on the last iteration of each step. The degradation of the results in the back-compatibility test is no surprise. Another test that has not been performed here would be to operate the sequence of assimilation systems in the reverse order and compare the final parameters and validation results. The computing effort is the same as for the order presented here.

We agree with the reviewer that with the Gaussian framework it would have been optimal to propagate the full error covariance matrix. We however did not propagate the off-diagonal terms for the following reasons:

• It was a substantial simplification in term of system engineering to only propagate the diagonal terms when we initially built the system.

• The error covariance terms were not large as we obtained correlations that were on average below 0.3.

• Propagating the variances appears to be "sufficient". This is indirectly verified given that the back-compatibility is achieved to a very good level on average (figure 8). The degradation of the fit is indeed marginal: i) for FluxNet data the change of RMSE between step 2 and step 3 is negligible compare to the improvement achieved during step 2; ii) for NDVI, the change is only significant for Temperate deciduous tree and C3 grasses but the RMSE in step 3 is still much lower than with the prior parameter set. These back-compatibility tests thus indicate that the information provided in previous steps is not lost during subsequent steps.

• Overall, we started this study with only the propagation of the variances from one step to the next, but we also investigated the impact of not propagating the covariance

with simpler models and set up. MacBean et al. (2016, in review) analyzed these issues: their main finding is that not propagating the covariance terms is likely to have a small influence on the posterior parameter values. Note finally that since submitting this paper we are working on improving the system to propagate the diagonal terms.

Our approach provides thus a simple step-wise framework that is able to account for the three sources of information with no significant lost of information from one step to the next as revealed in figure 8 and with coherent parameter changes.

However, we acknowledge that this issue was probably not highlighted enough in the original text. We have thus slightly reinforced it in the discussion section and we also mention it in the method section.

The assimilation of a statistical index, i.e. NDVI, is somewhat beyond state of the art, as assimilation of the related physical variable, FAPAR, has been demonstrated for multiple LSMs (Knorr et al., 2010, Schurmann et al, 2016). The required physical model of FAPAR is available in ORCHIDEE (Naudts et al., 2015).

We strongly disagree with this statement for the following reasons, which have been outlined in Bacour et al. (2015) and MacBean et al. (2015) but we chose not to repeat in this paper so as not to have overlap between the two studies:

• Studies have shown that considerable discrepancies exist between so-called "high-level" satellite products such as LAI or fAPAR, especially when considering their magnitude (D'Odorico et al., 2014; Garrigues et al., 2008; Pickett-Heaps et al., 2014). These differences / uncertainties are attributed to differences in the processing chains, in particular the radiative transfer models that are used to derive these products (based on different physics and assumptions) Figure 1 below, taken from D'Odorico et al. (2014), illustrates the issues with 3 state of the art FAPAR products. The maps highlight the differences in space over Europe, while the frequency distributions for July (at the peak of the growing season) are clearly significantly different between these products.

• We have therefore considered a vegetation greenness index, the Normalized Difference Vegetation Index (NDVI) and we only used the temporal information brought by this product using normalized values. The impact of using raw fAPAR data on the optimized model parameters for ORCHIDEE has been detailed in Bacour et al. (2015). This study shows that the maximum fAPAR values during the peak of the growing season imposes strong constraint on the maximum photosynthetic capacity parameters (VCMAX, VJMAX) which could lead to the estimation of spurious parameter values. Similar results have been obtained by Zobitz et al. (2014) who showed that the assimilation of FAPAR data (alone) could result in unrealistic simulated NEE values.

• Given that fAPAR and NDVI are nearly linearly related and that we normalize the signal between 0 and 1, using one or the other variables is thus equivalent.

• Note finally that Schürmann et al., (2016) obtain a very large impact on the gross and net carbon fluxes with the assimilation of raw fAPAR data; they could not evaluate if it degrades or improves the maximum photosynthetic uptake (at least it pulled the GPP towards values much lower than the data-driven product of Jung et al. (2011) based on FluxNet data).

Overall, we appreciate that we have not discussed enough our choice, given our wish to limit the overlap with MacBean et al. (2015) as mentioned above. So we have now added one sentence in section 2.4.1 to justify more clearly our choice: "Given that considerable discrepancies exist between so-called "high-level" satellite products such as LAI or fAPAR regarding their magnitude (D'Odorico et al., 2014), we thus only use the temporal information in the NDVI observations and normalized both the model FAPAR output and the NDVI observations to their 5th and 95th percentiles (following Bacour et al. (2015))."

Note finally that Naudts et al. (2015) describe a version of ORCHIDEE, named ORCHIDEE-CAN, that was not available at the beginning of the study and that has only been validated for European ecosystems (i.e. not the tropical ones for instance).

Specific Comments:

p11: 184 parameters is misleading, as none of the three systems estimates that many parameters Why are KsoilC parameters differentiated per region?

We agree that 184 is the total number of parameters optimized, but that in step 2 and step 3 the number is slightly lower and that in step 1 it is indeed much lower. We have corrected the text to be more precise.

As for the KsoilC parameters, they scale the initial values (after spin-up) of the modeled slow and passive soil carbon pool sizes, in order to take account of all the historical effects not accounted for in the model that would result in a disequilibrium of these pools in reality. It would thus be a strong hypothesis to assume that the "historical effects" impacted the soil carbon content uniformly. Indeed the history of land cover changes and land management largely differ between region/ecosystems and not accounting properly for their impact on soil carbon stock is a crucial point to address. For the global scale optimization step, we used 30 KsoilC,reg parameters corresponding to 30 regions (see Fig. A2). Rayner et al. (2005) used a similar approach with 13 coefficients for their 13 PFTs. In our case we choose to define the region not on a PFT basis but following large ecosystems regions that could be coherent for the history of land cover change, land management as well as ecosystem and edaphic conditions. Note that Schurmann et al. (2016) use only one global scalar and recognize that this is one of the major limitations of their approach. However, we acknowledge that the choice of 30 regions was not enough justified and we thus added one sentence in section 2.3.3: "For the global scale optimization step, we used 30 KsoilC,reg parameters corresponding to 30 regions potentially coherent for land use and land management history as well as ecosystem and edaphic properties (see Fig. A2)."

p23: Why are the FLUXNET assimilations performed per site and not simultaneously? How is the error of the parameter averaged over PFTs calculated.

We guess the reviewer is asking why the assimilations are performed per PFT and

not simultaneously for all PFTs. The reason was technical as doing it per PFT was slightly simpler and it allowed us to made several tests independently for each PFT. This allowed running smaller "optimization runs" in terms of requested memory and computing time, which proved to be more efficient given some random system failure (due regularly to failure in disk access).

As a drawback, we indeed had to average the estimated values for few global parameters (not dependent of the PFT). For the uncertainty associated to these parameters we averaged the variances. We have thus improved the text to describe more precisely the treatment of the error for these parameters.

Eq.(1) in the manuscript does not correspond with Eq. (1) in Tarantola (1987).

We agree that this was a mistake and drop the reference to Eq. (1) in Tarantola (1987) and replaced by Chapter 4 (where least square problems are described).

p21: After assimilation of atmospheric CO2 it is no surprise that the trend is close to observations.

We agree that this is probably the strongest constraint in the optimization and that it is clearly expected that we match the atmospheric CO2 trend with the optimization of a large set of parameters. We nevertheless kept the sentence but added at the end the term: "as expected".

p24: Fluxes are calculated from 2000 to 2009. Why are concentrations in Figure 6 not shown over the same time span?

We have shown in figure 6 only the time period when the atmospheric concentrations are used in the optimization. For the fluxes, given that we wanted to compare with other approaches, such as the GCP estimates, we have run the optimized ORCHIDEE model over a longer period to provide an mean estimate over the 2000 decade. Note finally that restricting the period in figure 6 to three years also help to see more clearly the improvements in term of seasonal cycle.

p41: 36 regions while in text it is 30.

It was a mistake. Corrected

Technical Corrections:

p e l 16: "remains"change to "remain"

Corrected

p 13 l 13: "we did not propagated" change to "we did not propagate"

Corrected

p 9 l 12: "et al., (1980)" change to "et al. (1980)"

Corrected

References cited:

Kaminski T, Knorr W, Rayner P, Heimann M: Assimilating atmospheric data into a terrestrial biosphere model: A case study of the seasonal cycle. Global Biogeochem Cycle 2002, 16:1066 doi: 10.1029/2001GB001463.

Knorr, W., and J.-P. Schulz (2001), Using satellite data assimilation to inferglobal soil moisture and vegetation feedback to climate, in Remote Sensing and Climate Modeling: Synergies and Limitations, edited by M. Beniston and M. Verstraete, pp. 273– 306, Springer, New York.

Knorr, W., T. Kaminski, M. Scholze, N. Gobron, B. Pinty, R. Giering, and P. P. Mathieu (2010), Carbon cycle data assimilation with a generic phenology model, Journal of Geophysical Research-Biogeosciences, 115.

Rayner, P. J., M. Scholze, W. Knorr, T. Kaminski, R. Giering and H. Widmann (2005), Two decades of terrestrial carbon fluxes from a carbon cycle data assimilation system (CCDAS), 19, doi:10.1029/2004GB002254.

Rayner, P. (2010) The current state of carbon-cycle data assimilation, Current Opinion in Environmental Sustainability, 2, 289-296.

Raoult, N. M., Jupp, T. E., Cox, P. M., and Luke, C. M.: Land surface parameter optimization through data assimilation: the adJULES system, Geosci. Model Dev. Discuss., doi:10.5194/gmd-2015-281, in review, 2016.

Schurmann, G. J., Kaminski, T., Kostler, C., Carvalhais, N., Voßbeck, M., Kattge, J., Giering, R., Rodenbeck, C., Heimann, M., and Zaehle, S.: Constraining a land surface model with multiple observations by application of the MPI-Carbon Cycle Data Assimilation System, Geosci. Model Dev. Discuss., doi:10.5194/gmd-2015-263, in review, 2016.

Wang YP, Leuning R, Cleugh H, Coppin PA: Parameter estimation in surface exchange models using non-linear inversion: How many parameters can we estimate and which measurements are most useful? Global Change Biol 2001, 7:495-510.

Zobitz, J. M., D. J. P. Moore, T. Quaife, B. H. Braswell, A. Bergeson, J. A. Anthony, and R. K. Monson (2014), Joint data assimilation of satellite reflectance and net ecosystem exchange data constrains ecosystem carbon fluxes at a high-elevation subalpine forest, Agric. For. Meteorol., 195–196, 73–88.
* * *
[Figure]

**Fig. 1.** Figure 1 (from D'Odorico et al., 2014): Left: Maps of FAPAR from TIP, MGVI and MCD products (a–c), their differences (d–f), and their correlations (g–i). Temporal resolution: a–f) July monthly composi

---

## Author Response (AR1)

**Response to reviewers**

We have thoroughly responded to the four reviewer's comments in four specific files. We believe we have addressed all comments of the reviewers and we thus have uploaded a final version of the paper accounting for the suggested changes they have made.

Best Regards,

Philippe Peylin

[revised manuscript text omitted]

---

## Author Response (AR2)

**Response: second round**

**Editor's comments**

Thanks for submitting your revised manuscript. In reviewing your responses, especially reviewer #4, you have provided strong rebuttals to the reviewer, however, you have not made any substantial changes to the manuscript to clarify the various points raised by the reviewer. I ask that you and your co-authors please re-consider All the reviewer comments (not just RC4), and make sure enough changes are made to the manuscript to clarify the criticisms made the reviewers, rather than only address this in your reply. This will help future readers of the manuscript who may have similar questions. For example, i would expect to see a few sentences in your introduction, which explains how your work further builds on the Rayner, Knorr and Wang papers, as per your rebuttal.

Response:
We thank the editor for his comment and acknowledge that the changes we have made to the text could have been slightly more substantial to highlight the issues raised the 4 reviewers. We thus report below the 4 reviews with our initial responses and describe for each comment any new additional changes we have made to the main text (when relevant). We hope that with these additional revisions ensure that all concerns raised initially by the reviewers are properly discussed in the manuscript.

Note that in our rebuttal to the comments of reviewer 4, we made a little mistake when citing Wang paper as it should have been Knorr and Heimann (1995). Wang et al. (2001) was already cited when describing studies that have assimilated in situ FluxNet data.

**Response to reviewer 1**

*Peylin and colleagues describe a Carbon Cycle Data Assimilation System based on the land-surface model ORCHIDEE optimized against NDVI data, eddy covariance CO2 flux data, and atmospheric CO2 data. For practical reasons, these three data streams are used successively in three steps. The paper describes the system, assesses its performance (especially the self-consistency across the three steps), and some features of the resulting carbon cycle fluxes and stocks. The authors conclude that the ORCHIDEE land-surface model is now structurally adequate enough to bridge the information from the three data streams, though they also highlight further steps that need to be taken to represent the global carbon cycle more accurately.*

*The study represents an interesting and relevant development in the understanding of the carbon cycle consistent with available data. There are open issues (for example the short assimilation period precluding various processes to be constrained and assessed) but these are clearly acknowledged in the paper. I find the presentation clear and convincing. In my opinion, the work should be published in Geoscientific Model Development.*

*First response:*
We thank the reviewer for his positive appreciation of the manuscript.

*p 5 l 6: The associativity is true for linear systems, but is it really also true for nonlinear systems? (I nevertheless agree to the arguments given in favor of the step-wise approach.)*

*First response:*
The associativity argument, detailed in Tarantola (2005), relies on the combination of probability density functions (PDFs) through the Bayes theorem without any assumption on the linearity or non-linearity of the system. It is a general property associated to the combination of probability distribution (i.e., the description of the probability of an event, based on conditions that might be related to that event). So the non-linearity does not invalidate the step-wise approach; it only highly complicates the computation of the full PDFs at each step and thus their propagation. The necessary simplifications that are made in this case lead to the complication investigated in the paper. In the case of a linear system, we could have easily calculated and propagated the full PDFs.
We have completed one sentence in the introduction to reinforce that the associativity does not depend on the linearity of the system:
"This multiplication is associative so it makes no differences whether it is performed in one step or several (and whether the system is linear or not)."

*Second response:*
No addition to the previous modification as we already added one sentence into the manuscript as detailed above.

*p 8 l 13: What is the uncertainty due to incomplete sampling of the diurnal cycle?*

*First response:*
We have not estimated the uncertainty due to a possible 20% gap. However, in order to compute the daily mean values, we have used gap-filled data so that the uncertainty only arises due to errors in the gap-filling procedure. Such uncertainty, usually less than 20 % (Lasslop et al. 2008), would thus only affect 20% at maximum of the diurnal cycle; it would thus have an overall negligible impact compared to the model uncertainty that need to be also accounted for.

*Second response:*
We completed the text with the following sentence: "Note that uncertainties due to incomplete sampling of the diurnal cycle are likely very small (less than 5%) as the error in the gap-filling procedure is usually less than 20% (Lasslop et al., 2008)."

*p 14 l 25-26: I was wondering whether the presence of step functions, creating discontinuities, still allows a well-defined solution of the minimization?*

*First response:*
We agree that step functions may complicate the minimization problem, creating potentially non-smooth cost function, i.e. with a "singular" point where the derivative is not defined and not continuous. However the use of finite differences to compute the gradient for these parameters allows defining a "mean derivative" at any point (mean as for non local). As a result the iterative approach (BFGS algorithm) used to search for the minimum may oscillate if we end up for a given parameter in the vicinity of the discontinuity, but it will not diverge. However such situation is very unlikely to occur and we have checked that no obvious oscillation for each parameter was occurring.
We completed one sentence in the text to precise this point:
"A finite difference approach was used for these parameters in order to define a mean derivative at any point".

*Second response:*
We completed the added sentence: "A finite difference approach was used for these parameters in order to define a mean derivative at any point; we also checked that no spurious oscillations occurred for these parameters during the minimization iterations."

p 17 l 25-30: Add references for "University of Stuttgart" and "ENSTO-E". Explain abbreviation "IER".

*First response:*
We now explain the "IER" abbreviation and we provide two web references for both "University of Stuttgart" and "ENSTO-E".

*Second response:*
No addition

*p 20 l 14: What does "conditions" mean here?*

*First response:*
We have changed the text to precise that "conditions" was referring to station sampling air "representative of different geographical regions of influence".

*Second response:*
No addition p 21 l 4: Clarify whether this is the prior before step 1 or before step 3?

*First response:*
It is the prior of step 3 and we have now clarified the text.

*Second response:*
No addition p 22 l 21: Can you give just a brief summary of the reasons here?

*First response:*
We have improved the text to provide a brief summary of these reasons:
- The NDVI or fAPAR mainly constrain the timing of the ecosystem GPP and only to a small extent the amplitude of the GPP, given that the satellite signal is likely to saturate during the peak of the growing season.
- The NDVI or fAPAR do not constrain the ecosystem respiration at all, which is an important component of the NEE, at least at seasonal to annual scales.

*Second response:*
We only slightly change the first reason as: "The NDVI or fAPAR mainly constrain the timing of the leaf phenology (and thus indirectly the GPP) and ….. "

p 25 l 24: These are clearly not the numbers shown in Fig 10 right.

*First response:*
We apologize for such mistake as the numbers correspond indeed from a previous experiment.
We have corrected the text with the exact numbers from figure 10b.
Note that the exact numbers from figure 10b do not change the overall message and the rather small reduction of the GPP from the prior to the posterior of step 3.

*Second response:*
No addition p 27 l 6: Can you explain (here or earlier) why you used individual grid points rather than the whole grid?

*First response:*
The main reasons for choosing only a set of individual grid points are twofold:
- First, we proposed to used only the model grid point that are covered by a vegetation fraction greater than 60% for a given PFT, avoiding the use of grid points with a large mix of PFTs so that the optimization of the phenological parameters for each PFT is more straightforward. This is explained in section 2.4.1 (step 1).
- We then limited the selected set of points to 15 in order to significantly reduce the computing time of the step 1 optimization. Doing so further allows evaluating the optimized model at all pixels that were not used for the optimization.

We added a sentence in section 2.4.1 (step 1 paragraph) to better explain our choice:
"Only grid cells that included vegetation fraction of greater than 60% for the PFT optimized were considered. We selected a set of grid points instead of the whole grid to substantially decrease the computing time; but the remaining points are used for the evaluation of the optimized model."

*Second response:*
No addition p 29 l 10: I think you should also mention the errors in the prescribed fossil fuel and ocean fluxes.

*First response:*

We added at the end of the paragraph a sentence to mention the importance of the errors associated to the prescribed fossil fuel and ocean fluxes:
"Additionally, uncertainties on the other components of the carbon cycle, such as fossil fuel and biomass burning emissions and ocean fluxes, can be also critical when using atmospheric $CO_2$ as a constraint."

*Second response:*
No addition

Fig 5: I don't see any grey lines.

*First response:*
We have corrected the caption as we do not included the "prior of step 1" in this version of the figure in order not to overload the figure with too many curves.

*Second response:*
No addition

**Response to reviewer 2**

*Peylin et al. contribute an interesting study on the effect of using a stepwise optimisation rather than merging all data streams in a single cost function. This is a procedure that we have used in e.g. catchment scale water quality modelling where one first calibrates the water cycle before calibrating parameters relevant to nutrient diffusion (e.g. Exbrayat et al., 2011). However, this approach has not been investigated in details in the frame of the (global) carbon cycle. Therefore, I agree with reviewer #1 that this paper is highly relevant to the community. I particularly like the several steps used by the authors to reconcile site-scale calibration with global atmospheric concentrations.*

*I have some very minor comments on the paper that should be straightforward to address:*

*First response:*
We thank the reviewer for his positive evaluation of the paper.

p.5 l.1: another alternative to stepwise optimisation and simultaneous procedure is a simultaneous, multi-objective approach based on the Pareto ranking of several cost functions to account for trade-offs (e.g. Yapo et al., 1998). Would it be realistic to use such an approach in this system to avoid the increase in RMSE against MODIS NDVI from step1 to step2 (fig. 8, TeBD)?

*First response:*
The suggestion of using a simultaneous multi-objective approach based on the Pareto ranking of several cost functions to account for trade-offs as in Yapo et al. (1998) could indeed appear as an alternative approach. However, it is based on a random generation of parameter sets in order to find the so-called "Pareto parameter space" where all parameter sets lead to equal overall objective function but with trade-offs between individual cost functions. With the global carbon cycle optimization problem and the ORCHIDEE model the approach in Yapo et al. (1998) is from a computational point of view not feasible at all. For a specific hydrology problem with 13 parameters to be optimized and two cost functions the proposed algorithm required more than 68000 function evaluations to converge to a solution. Given that our total number of parameters is on the order of 100 and that we can not afford a large number of global simulation for the evaluation of the cost function linked to atmospheric CO2 data (few tens to a hundred), such method is thus clearly not suitable.

Finally we should mention that the increase in RMSE against MODIS NDVI from step1 to step2 is relatively small compared to the initial improvement of the RMSE during step1 optimization.

*Second response:*
We have included in the introduction a new sentence to mention that such alternatives are not affordable: "Note finally that more complex approaches based on random generation of parameter sets, such as the multi-objective approach using the Pareto ranking of several cost functions (e.g. Yapo et al., 1998), are not yet affordable for global LSMs from a computational point of view."

p.5 l.12-13: Using a restricted number of parameters is a valid point but it needs to be mentioned here that one must proceed to some sort of sensitivity analysis to accurately select these parameters.

*First response:*
We agree that the selection of parameters should follow a rigorous sensitivity analysis, using for example the MORRIS algorithm (global sensitivity analysis; Morris, 1991). In our case, we have done such sensitivity analysis in previous studies with the ORCHIDEE model; we have thus kept the same set of parameters that was identified in these studies.
We have improved the text adding:  "(following a global sensitivity analysis)".

*Second response:*
No addition p.8 l.1: why not using only days with data?

*First response:*
We agree with the reviewer that we could have used only the days with data. However, for practical implementation it was easier to interpolate the time series that have sparse data. We have checked however that using this interpolation does not change the results of the optimization.

*Second response:*
We completed the text adding that such choice was made for "practical implementation"

p.9 l.26: please mention the resolution of the model here

*First response:*
We added the resolution of the LMDz model.

*Second response:*
No addition p. 10 l.1: is it robust to assume that carbon pools are at equilibrium in 1990? Could this system use a prior from soil and biomass maps instead (like e.g. Bloom et al., 2016)?

*First response:*
The choice to bring the model at equilibrium for the carbon pools in 1990 is a compromise. We do not think that our optimization system could easily use soil map (such as the HWSD data set) or forest biomass estimates (such as the Saachi et al (2011) map for the Tropics).

For the soil carbon content, it is not straightforward to use the HWSD map to force a global model like ORCHIDEE. First, with the "CENTURY" soil carbon model used in ORCHIDEE, the turn over time of the soil organic matter (for each reservoirs) together with the rate of organic matter input to the soil (litter and root turnover) determine a total soil carbon content that is in balance with all components of the model. It is thus difficult to optimize the soil carbon content with global estimates such as HWSD, while keeping the internal model consistency. One way would be to optimize the parameters controlling the turnover times as well as the soil carbon input over a long spin up period (several 1000 years); this is currently not feasible with the optimization of several parameters, especially without the adjoint of the model. Second the model does not represent yet high soil carbon content such as peat land or permafrost, while these soil type are usually taken into account in the observations. Finally, the HWSD soil C map corresponds primarily to the carbon content from 0 to 1 meter of soil; we would need first to adjust the observation so that it matches the total soil carbon content that is modeled. Overall, it is a rather complex process to optimize the soil carbon content in the case of ORCHIDEE if we want to keep the internal coherence of the model to improve its predictive skill. Ongoing works are occurring at LSCE to determine the best approach to assimilate soil carbon content observations or potentially turnover time of soil organic matter (derived from observations) into ORCHIDEE.

Similar issues also pertain to the assimilation of forest biomass maps (such as Saatchi et al. 2011). We cannot change easily the above ground biomass in ORCHIDEE without violating the overall carbon allocation scheme and the internal consistency of the model (i.e., the ratio of carbon content between the different reservoirs). We thus would need to perform an optimization of the parameters controlling carbon allocation as well as the input of carbon over a period corresponding at least to the age of the forest. This is a difficult task that was investigated in Thum et al. (in revision) for site scale observations.

The study of Bloom et al. (2016) is slightly different in the sense that the model is less complex than ORCHIDEE and that they could do an optimization over long time period, as needed for carbon stock observations. However, we acknowledge that using carbon stock observations is the next challenge as already mentioned it in the discussion section.
For the soil carbon content we thus choose a compromise which is to bring the model to equilibrium in 1990, perform a transient simulation over 10 years and then optimize an ensemble of coefficients that scale (either at site level or globally for several regions) the initial soil carbon content (the slow and passive pools) in order to account for the all past effect that led to deviation from the "equilibrium assumption".

*Second response:*
We have improved the text to mention the issues and difficulties to use soil carbon and forest biomass maps with the ORCHIDEE model in order to avoid the classical spin-up approach. In section 2.3.1 we added the following text (that also refers to the discussion in section 4):

"Note that the use of soil carbon data, such as from the Harmonized World Soil Database (as well as above ground biomass data), to initialize the model is not straightforward and represents a challenge to keep the internal model consistency, given that the three soil carbon reservoirs of the CENTURY model are in balance with all components of the model, in particular the input through the different litter pools. Computational and scientific issues to avoid a spin-up approach are still under investigation with ORCHIDEE (see discussion section)."

p. 18 l.6: How is fire simulated during spin-up?

*First response:*
During spin-up the ORCHIDEE model is run with fires provided by a "generic" fire emission module that is thus not based on the satellite observation of fire occurrence and fire extension. Such simplification is necessary, as we do know have the past history of fire occurrence to guide the spin-up procedure. We have clarified in the text that the inclusion of GFEDv3 fire emission is not used for the model spin-up.

*Second response:*
No addition p.26 l.29: perhaps "ecosystem data streams" is more correct (LE is not carbon sensu stricto)

*First response:*
Indeed LE is not a carbon cycle data stream but it is related to the carbon cycle through the stomatal conductance at the leaf level. In order to keep the word "carbon" we have change the expression to "carbon cycle – related data streams"

*Second response:*
No addition p. 29 l.17: see also Bloom et al. (2016)

*First response:*
We added the reference "Bloom et al. (2016)"

*Second response:*
No addition

**Response to reviewer 3**

**General comments :**
The manuscript reports development and application of a data assimilation system which is used to produce a version of ORCHIDEE model optimized to reproduce NDVI, net ecosystem exchange and latent heat flux at land validation/flux tower sites and CO2 seasonality at background CO2 monitoring sites. The stepwise optimization approach is proposed as a simplified alternative to optimizing model to fit NDVI, flux tower data and atmospheric CO2 data simultaneously. Despite splitting the process in several stages authors succeeded to find a set of parameters allowing the model to fit all types of constraints.

The manuscript is well written, and presents an original and valuable contribution. It can be published after minor revision, hopefully addressing the comments listed below.

*First response:*
We thank the reviewer for his positive review and we describe below how we have address his comments.

General comment 1. In the optimization framework adopted by authors, model parameters optimized at the set of flux tower sites are later extrapolated to whole land surface using available spatial data on vegetation type, weather and soil type information as drivers. The flux tower site optimization is made by combining several sites within same vegetation type in one group, and average flux seasonality is shown to be improved by the optimization. The variability of the fluxes due to soil quality and slope/drainage within same vegetation type is not directly captured by this approach, while some studies (Ise an Sato, 2008) suggest there is a way to address site level differences in productivity potential (edaphic variability) based on remote sensing data. It would be relevant to mention this factor in discussing reasons for remaining spread in the degree of success that can be achieved using one set of model parameters for optimizing fluxes at several sites of same vegetation type.

*First response:*
We agree that the optimization approach is not able to account for all sources of variability for the carbon and water fluxes measured at FluxNet sites and in particular those linked to edaphic conditions (soil quality, slope and drainage,…). This is indeed a potentially important limitation of current global LSM. The study of Ise and Sato (2008) brings an interesting perspective to include part of the edaphic variability although it relies on strong hypothesis: i.e., the impact of edaphic variability directly controls the vegetation distribution at high spatial resolution and the GLC2000 land cover product is able to capture the differences between high, medium and low productivity ecosystems. The authors have shown that taking into account the spatial variability of the land cover from GLC2000 could significantly improve the model simulation of Leaf Area Index at high latitude in North America, but not really over Siberia.
We thus believe that it is an interesting direction of research, worth to be mentioned in the discussion, but not specific to data assimilation as it concerns the global LSM performance in general. We thus only added in section 4 the following sentence:
"Finally, one can mention new approaches based on remote sensing data to account for site level differences in productivity potential due to edaphic variability (soil quality and slope/drainage) within the same vegetation type (Ise an Sato, 2008)."

*Second response:*
We slightly revised the sentence added initially to mention that Ise and Sato (2008) only illustrated the potential for high latitude in North America:
"Finally, one can mention new approaches based on remote sensing data to account for site level differences in productivity potential due to edaphic variability (soil quality and slope/drainage) within the same vegetation type, as illustrated for high latitudes in North America (Ise and Sato, 2008)."

**Detailed comments**

Page 01 – Line 03. In addition to "incorrect model parameter values" one should mention uncertainty in spatial distribution of the parameters coming from the maps of soil properties, topographic features, vegetation types.

*First response:*
We agree and have added: "poor description of land surface heterogeneity (soil and vegetation properties),…"

*Second response:*
No addition

Page 04 – Line 05. Randerson et al (1996) paper can be mentioned among influential studies that use air concentration as constraint

*First response:*
We agree that Randerson et al. (1996) was a key paper that used atmospheric $CO_2$ concentration as a constraint to evaluate the impact of specific processes linked to organic matter decomposition. However, this study only used the data to evaluate the model output but not to constraint some parameters with a formal optimization procedure. Given that the introduction is focusing on studies that have used an optimization procedure, we prefer not include Randerson et al. (1996); else we would need to include several other studies that have similarly used atmospheric observations to validate specific ecosystem processes.

*Second response:*
No addition

Page 04 – Line 26 To extend a list studies using multiple input streams and C stock data in assimilation (Saito et al 2014) can be added.

*First response:*
We agree that this study was missing from the list of data assimilation studies and we thus added it.

*Second response:*
No addition

Page 20 Line 13. In many transport models it appears difficult to match $CO_2$ seasonal cycles in PBL and free troposphere at the same time, which can be attributed to simulated PBL height biases and biases in other processes. The problem can lead to finding a set of model parameters that are optimized well for LMDz model with its PBL height and PBL ventilation rate, but not performing that well when model is different. It would be useful to add figure showing match with free tropospheric data for model validation. Aircraft data and TCCON data can be used for validation, especially high latitude sites know for high seasonal amplitude such as Poker Flats Alaska, or TCCON at Sodankula (Lindqvist et al., 2015). A useful check would include use of vertically integrated profile data at airborne observation sites (Nakatsuka and Maksyutov, 2009), as it is more stable against the PBL height biases.

*First response:*
We agree with the reviewer that the representation of the PBL spatial and temporal dynamic is crucial when using atmospheric $CO_2$ data to optimize ecosystem model parameters or surface fluxes. However, using few scarce free tropospheric data to evaluate the performance of the optimization will not bring significant information as the dynamic of the PBL varies substantially between regions; we thus would need to have a large set of free tropospheric data to provide a comprehensive validation.
The second suggestion of the reviewer concerns vertically integrated data, such as TCCON. Indeed they depend less on PBL height biases, but they however depend crucially on other large-scale uncertain model features, such as the north – south overall mixing.
Finally a technical constraint complicated the evaluation of the model output with these new observations. We indeed used pre-calculated transport fields (model Jacobian) that were calculated at a selected set of surface stations (computationally intensive process) with an older version of LMDz: version 3. It would thus require us to run again the LMDz transport model version 3, which was not feasible in a reasonable time frame period.

Given this technical constraint and the relatively small-expected gain from the evaluation at free tropospheric sites or at TCCON sites, we propose instead to use additional surface stations not used during the optimization (but where the Jacobian are available). We have used 17 sites that are more representative of continental fluxes than the stations that were assimilated and 7 sites that correspond to Pacific Ocean cruises that were left aside in order not to overweight that particular region in the optimization. This independent atmospheric $CO_2$ evaluation illustrates that the improvement is not only valid at the optimization sites. On average the mean RMSE for the 27 additional sites is 10.5 ppm for the prior of step 1 (prior of ORCHIDEE), 2.8 ppm for the prior or step 3 and 2.1 ppm for the posterior of step 3. The corresponding values for the 53 sites used for the optimization are: 10.5, 2.45, 1.8 ppm, respectively. The error reduction during step 3 is thus similar for both the assimilated and the validation data sets. We added this additional evaluation in the paper (section 3.1.3) as an independent validation of the optimization with selected atmospheric $CO_2$ observations.

*Second response:*
We modified the text in order to mention why we have not used aircraft data and TCCON data for validation, even if these free tropospheric data are less dependent on the PBL height. Also when mentioning the use of additional surface stations for validation we now justify why we have not used these particular sites in the optimization.
The text at the end of section 2.2.3 now becomes:
"We also used additional sites to evaluate the result of the optimization (locations indicated in Fig. 3): this included17 continental sites that are more directly influenced by local fluxes potentially not well captured at the considered LMDz spatial resolution and 7 sites from Pacific Ocean cruises that were not included in the optimization in order not to overweight the data contribution from that particular region. Note that we did not consider free troposphere aircraft data or column integrated measurements (TCCON sites) in this evaluation, although they are less sensitive to biases in the Planetary Boundary Layer representation, given that i) we are using pre-calculated transport fields previously computed at surface stations only, and ii) few scarce free tropospheric datasets will not bring much more information to the additional surface stations."

**Response to reviewer 4**

**General comments:**
The manuscript presents a sequence of parameter estimation exercises for the ORCHIDEE Land Surface Model using a CCDAS data assimilation framework. Firstly, NDVI data are assimilated at point scale. Secondly, FLUXNET data are assimilated at point scale. Thirdly, atmospheric CO2 data are assimilated at global scale. The presentation of the material is excellent, despite some minor inconsistencies.

*First response:*
We thank the reviewer for having commented our manuscript. We explain below why we disagree to some of his major comments.

The novelty of the material is limited. What the authors present as a step-wise system, are in fact three systems that are operated in a sequence. The interface between these systems is minimal: It consist of selected parameters with error bars but excluding the error covariance that are passed in one direction. The step-wise approach is not new. It is desribed, for example, by Rayner et al. (2005): They assimilate NDVI in the first step and atmospheric CO2 in the second step. The system for assimilation of NDVI is described in more detail elsewhere (MacBean et al., 2015). The system for assimilation of FLUXNET data is described in more detail elsewhere (Kuppel et al., 2012, 2014). What is left is the system for assimilation of a single data stream, i.e. the atmospheric CO2 data from 2002 to 2004. The description of the assimilation method is provided elsewhere (see above references). The ORCHIDEE LSM, the LMDz CTM and the use of influence functions was also described elsewhere (see references in section 2.3.2). The assimilation of atmospheric CO2 using a combination of an LSM and a CTM and prescribed emissions from other components of the carbon cycle is not new either. It was presented by Rayner et al. (2005) and applied for a time span of two decades. In summary the manuscript is not suitable for GMD because it fails to present "substantial new concepts, ideas, or methods".

*First response:*
We disagree with the main criticism that our study does not provide new ideas or methods. In order to explain this further we need to lay out the evolving state of carbon cycle data assimilation.

Systems that apply the well-established methods of data assimilation to models of the carbon cycle at various scales have been around for nearly two decades. Wang et al. (2001) and Kaminsky et al (2002) antecede Rayner et al. (2005). The problem pointed out by Rayner et al. (2010) is that information was not transferable between either different sites or different datastreams. Rayner et al. (2005) for example, did not expose any of the phenological parameters of the assimilation from Knorr et al. (2001) in their assimilation so there could be no test of consistency. Rayner et al. (2010) pointed out that evolution of both models and methods was necessary for comprehensive assimilation. The current paper represents an important step in that evolution though by no means the final one.

- The paper describes for the first time (to our knowledge) a system that is able to assimilate three major carbon cycle data streams (vegetation activity from satellite, FluxNet data and atmospheric CO2) in a process-based land surface model used as the land component of an Earth System Model (ESM). No such system has been described so far, although this is a major challenge given the differences obtained for the carbon cycle in the last collection of model used for CMIP5 exercise (last IPCC report).

- The reviewer slightly overstates the achievement of Rayner et al. (2005). Although it did use soil moisture and radiation fields from an earlier assimilation from a related model (a simpler version) this was irrelevant to the narrative of that paper. The fields could just as easily have come from a direct satellite product. There was little comment made on the consistency between the two assimilations and no parameters passed between them. Since then, numerous global scale carbon cycle data assimilation studies have been published (above 10) all of them contributing additional aspects to these systems. However in those 10 years a third and crucial data stream, namely the widely used FLUXNET network of net CO2 and latent heat flux observations, has not yet been included with FAPAR and CO2 data in a global scale assimilation. All other studies to our knowledge use FAPAR data and CO2 or FLUXNET data, but none have used all three.

- The focus of the paper lies in the combination of these three data streams while the individual papers MacBean et al. 20015 and Kuppel et al. 2012, 2014 focus on the impact of each individual data stream. The major result of the paper is that for the first time "a state of the art global land surface model part of an ESM is able to capture with a reasonable accuracy the information content of three major data streams following a data assimilation procedure". This is no insignificant feat as it opens new perspectives to reduce the spread of the land carbon sink simulated by the CMIP5 suite of models and thus to reduce possibly the uncertainty in long-term climate predictions.

- We acknowledge that the chosen approach may not be the optimal one in a statistical sense with only the propagation of the error variance of the optimized parameters (see more detail in the response to the next comment).

- We and the reviewer agree that this is, fortunately, a burgeoning field of activity. Raoult et al (2016) and Schurman et al. (2016) are contemporarily studies, that rely on land surface models of an ESM. They present different strengths and weaknesses from this paper. However we are confident that the paper does represent an advance in available methods. Raoult, et al. (2016) only uses FluxNet observations to optimize the parameters of the JULES model, while Schürmann et al. (in revision) only assimilate two data streams (fAPAR and CO2) in JSBACH model (at coarse resolution, 10° x 10°). Note finally that the level of complexity of the ecosystem model is part of the problem: achieving an optimization with a simpler model does not guaranty that the framework would work with a more complex one.

Overall, the paper relies on old data assimilation concepts (published way before Rayner et al. 2005) but provides a new implementation (3 data streams with a state of the art component of an ESM) and opens the road for improved carbon – climate coupled simulations and improved climate predictions.

We acknowledge that we have not done enough to highlight the new features of this study, and thus we have emphasized these points in the "introduction" and "discussion and conclusion" sections.

*Second response:*
We agree that we have not included enough of the text in the above argumentation in the manuscript to justify the novel aspects of our study; we have thus further changed the introduction to emphasize the context so that the reader can sea more clearly why this paper represents an important step in the evolution of carbon cycle data assimilation systems. Several changes occurred in different places in the introduction in order to keep the initial logic:

- We first included the study by Kaminski et al. (2002) after the reference to Knorr et al. (1995), adding the sentence:
"Later, Kaminski et al (2002) constrain the seasonal cycle of SDBM with the same data stream."
- Then after the introduction of Rayner et al. (2005), we completed the description of this important first CCDAS study with the following sentence:
"Note that although Rayner et al. (2005) did use, in addition to atmospheric $CO_2$ data, soil moisture and radiation fields from an earlier assimilation from a simpler model version, no parameters were passed between the two assimilations and very little comment was made on the consistency between the two assimilations, an important issue that will be central to this paper."
- Finally, after the introduction of the main objective of our study, we added few sentences to reinforce the novel aspects:
"This study is the first (to our knowledge) to assimilate these three major data streams in a process-based LSM used as the land component of an Earth System Model (ESM), the French Institut Pierre Simon Laplace ESM. Two contemporary studies also optimize the parameters of the land component of an ESM; however Raoult, et al. (2016) only uses FluxNet observations to optimize the parameters of the JULES model, while Schürmann et al. (2016) only assimilate two data streams (fAPAR and $CO_2$) in the JSBACH model at coarse resolution (10° x 10°). Note finally that the level of complexity of the ecosystem model (and the spatial resolution) is part of the problem: achieving an optimization with a given model does not guarantee that the framework would work with a more complex or different one."

The scientific approach of passing reduced information on the parameters from one assimilation system to the next is questionable. The reviewer agrees with the author's statement: "It is important to note that this is an implementation question. Tarantola (2005) recasts the fundamentals of the approach as the conjunction or multiplication of probability densities. This multiplication is associative so it makes no difference whether it is performed in one step or several." However, an implementation of such a step-wise procedure needs to propagate the full Probably Density Function from one step to the next. In the Gaussian framework selected here this requires to propagate the full error covariance matrix and not just the error bars (see comment above). Error correlations are to be expected (see, e.g. Raoult et al., 2016). The change of the parameter space from one step to the next adds a further weakness as well as the dependence of **H**1 on the last iteration of each step. The degradation of the results in the back-compatibility test is no surprise. Another test that has not been performed here would be to operate the sequence of assimilation systems in the reverse order and compare the final parameters and validation results. The computing effort is the same as for the order presented here.

*First response:*
We agree with the reviewer that with the Gaussian framework it would have been optimal to propagate the full error covariance matrix. We however did not propagate the off-diagonal terms for the following reasons:

- It was a substantial simplification in term of system engineering to only propagate the diagonal terms when we initially built the system.

- The error covariance terms were not large as we obtained correlations that were on average below 0.3.

- Propagating the variances appears to be "sufficient". This is indirectly verified given that the back-compatibility is achieved to a very good level on average (figure 8). The degradation of the fit is indeed marginal: i) for FluxNet data the change of RMSE between step 2 and step 3 is negligible compare to the improvement achieved during step 2; ii) for NDVI, the change is only significant for Temperate deciduous tree and C3 grasses but the RMSE in step 3 is still much lower than with the prior parameter set. These back-compatibility tests thus indicate that the information provided in previous steps is not lost during subsequent steps.

- Overall, we started this study with only the propagation of the variances from one step to the next, but we also investigated the impact of not propagating the covariance with simpler models and set up. MacBean et al. (2016, in review) analyzed these issues: their main finding is that not propagating the covariance terms is likely to have a small influence on the posterior parameter values. Note finally that since submitting this paper we are working on improving the system to propagate the diagonal terms.

Our approach provides thus a simple step-wise framework that is able to account for the three sources of information with no significant lost of information from one step to the next as revealed in figure 8 and with coherent parameter changes.

However, we acknowledge that this issue was probably not highlighted enough in the original text. We have thus slightly reinforced it in the discussion section and we also mention it in the method section.

*Second response:*

We have slightly changed the method to justify our choice. The sentence is now: "Note that for simplicity we did not propagate the error correlations in this first implementation of the system, a simplification that appeared sufficient (see the consistency analysis in section 3.2); section 4 also discusses the potential impact of this simplification."
For the discussion section, we believe that it provides enough detail to the reader with a synthesis of the different points detailed in our first response to the reviewer (see above).

The assimilation of a statistical index, i.e. NDVI, is somewhat beyond state of the art, as assimilation of the related physical variable, FAPAR, has been demonstrated for multiple LSMs (Knorr et al., 2010, Schurmann et al, 2016). The required physical model of FAPAR is available in ORCHIDEE (Naudts et al., 2015).

We strongly disagree with this statement for the following reasons, which have been outlined in Bacour et al. (2015) and MacBean et al. (2015) but we chose not to repeat in this paper so as not to have overlap between the two studies:

- Studies have shown that considerable discrepancies exist between so-called "high-level" satellite products such as LAI or fAPAR, especially when considering their magnitude (D'Odorico et al., 2014; Garrigues et al., 2008; Pickett-Heaps et al., 2014). These differences / uncertainties are attributed to differences in the processing chains, in particular the radiative transfer models that are used to derive these products (based on different physics and assumptions) Figure 1 below, taken from D'Odorico et al. (2014), illustrates the issues with 3 state of the art FAPAR products. The maps highlight the differences in space over Europe, while the frequency distributions for July (at the peak of the growing season) are clearly significantly different between these products.

- We have therefore considered a vegetation greenness index, the Normalized Difference Vegetation Index (NDVI) and we only used the temporal information brought by this product using normalized values. The impact of using raw fAPAR data on the optimized model parameters for ORCHIDEE has been detailed in Bacour et al. (2015). This study shows that the maximum fAPAR values during the peak of the growing season imposes strong constraint on the maximum photosynthetic capacity parameters (VCMAX, VJMAX) which could lead to the estimation of spurious parameter values. Similar results have been obtained by Zobitz et al. (2014) who showed that the assimilation of FAPAR data (alone) could result in unrealistic simulated NEE values.

- Given that fAPAR and NDVI are nearly linearly related and that we normalize the signal between 0 and 1, using one or the other variables is thus equivalent.

- Note finally that Schürmann et al., (2016) obtain a very large impact on the gross and net carbon fluxes with the assimilation of raw fAPAR data; they could not evaluate if it degrades or improves the maximum photosynthetic uptake (at least it pulled the GPP towards values much lower than the data-driven product of Jung et al. (2011) based on FluxNet data).

[Figure]

*Figure 1 (from D'Odorico et al., 2014): Left: Maps of FAPAR from TIP, MGVI and MCD products (a–c), their differences (d–f), and their correlations (g–i). Temporal resolution: a–f) July monthly composite for the year 2011; g–i) July monthly composite for years 2009–2011. Spatial resolution (pixel size): a–c) 1 × 1 km; d–f) 3 × 3 km; g–i) 18 × 18 km.; Right: FAPAR frequency distributions for TIP, MGVI and MCD products over Europe for July of year 2011.*

Overall, we appreciate that we have not discussed enough our choice, given our wish to limit the overlap with MacBean et al. (2015) as mentioned above. So we have now added one sentence in section 2.4.1 to justify more clearly our choice:
*"Given that considerable discrepancies exist between so-called "high-level" satellite products such as LAI or fAPAR regarding their magnitude (D'Odorico et al., 2014), we thus only use the temporal information in the NDVI observations and normalized both the model FAPAR output and the NDVI observations to their 5th and 95th percentiles (following Bacour et al. (2015))."*

Note finally that Naudts et al. (2015) describe a version of ORCHIDEE, named ORCHIDEE-CAN, that was not available at the beginning of the study and that has only been validated for European ecosystems (i.e. not the tropical ones for instance).

*Second response:*
Given the risk of repeating the arguments outlined in Bacour et al. (2015) and MacBean et al. (2015) we have only slightly reinforce the justification mentioned in section 2.4.1. We added one sentence so that the overall justification becomes:
"Given that considerable discrepancies exist between so-called "high-level" satellite products such as LAI or fAPAR when considering their magnitude (D'Odorico et al., 2014), we thus only use the temporal information in the NDVI observations and normalized both the model FAPAR output and the NDVI observations to their 5th and 95th percentiles (following Bacour et al., 2015 and MacBean et al., 2015). Note that assimilating raw fAPAR data with the ORCHIDEE model led to a degradation of the NEE with the estimation of spurious parameter values (Bacour et al., 2015)."

Specific Comments:

p11: 184 parameters is misleading, as none of the three systems estimates that many parameters Why are **K**soilC parameters differentiated per region?

*First response:*
We agree that 184 is the total number of parameters optimized, but that in step 2 and step 3 the number is slightly lower and that in step 1 it is indeed much lower. We have corrected the text to be more precise.

As for the KsoilC parameters, they scale the initial values (after spin-up) of the modeled slow and passive soil carbon pool sizes, in order to take account of all the historical effects not accounted for in the model that would result in a disequilibrium of these pools in reality. It would thus be a strong hypothesis to assume that the "historical effects" impacted the soil carbon content uniformly. Indeed the history of land cover changes and land management largely differ between region/ecosystems and not accounting properly for their impact on soil carbon stock is a crucial point to address. For the global scale optimization step, we used 30 $K_{soilC,reg}$ parameters corresponding to 30 regions (see Fig. A2). Rayner et al. (2005) used a similar approach with 13 coefficients for their 13 PFTs. In our case we choose to define the region not on a PFT basis but following large ecosystems regions that could be coherent for the history of land cover change, land management as well as ecosystem and edaphic conditions. Note that Schurmann et al. (2016) use only one global scalar and recognize that this is one of the major limitations of their approach.
However, we acknowledge that the choice of 30 regions was not enough justified and we thus added one sentence in section 2.3.3: *"For the global scale optimization step, we used 30 $K_{soilC,reg}$ parameters corresponding to 30 regions potentially coherent for land use and land management history as well as ecosystem and edaphic properties (see Fig. A2)."*

*Second response:*
We did not further change the text of section 2.3.3 from the first response as scaling the initial soil carbon pools is a standard approach used is all carbon cycle data assimilation system so far.

p23: Why are the FLUXNET assimilations performed per site and not simultaneously? How is the error of the parameter averaged over PFTs calculated.

*First response:*
We guess the reviewer is asking why the assimilations are performed per PFT and not simultaneously for all PFTs, because all sites were included simultaneously for each PFT in a so-called "multi-site optimization". The reason for a per PFT optimization was technical as doing it per PFT was slightly simpler and it allowed us to made several tests independently for each PFT. This allowed running smaller "optimization runs" in terms of requested memory and computing time, which proved to be more efficient given some random system failure (due regularly to failure in disk access).

As a drawback, we indeed had to average the estimated values for few global parameters (not dependent of the PFT). For the uncertainty associated to these parameters we averaged the variances. We have thus improved the text to describe more precisely the treatment of the error for these parameters.

*Second response:*
We believe that the details provided in section 2.4.1- step 2 clearly answer the question raised by the reviewer.

Eq.(1) in the manuscript does not correspond with Eq. (1) in Tarantola (1987).

*First response:*
We agree that this was a mistake and drop the reference to Eq. (1) in Tarantola (1987) and replaced by Chapter 4 (where least square problems are described).

*Second response:*
No addition p21: After assimilation of atmospheric CO2 it is no surprise that the trend is close to observations.

*First response:*
We agree that this is probably the strongest constraint in the optimization and that it is clearly expected that we match the atmospheric CO2 trend with the optimization of a large set of parameters. We nevertheless kept the sentence but added at the end the term: "as expected".

*Second response:*
No addition p24: Fluxes are calculated from 2000 to 2009. Why are concentrations in Figure 6 not shown over the same time span?

*First response:*
We have shown in figure 6 only the time period when the atmospheric concentrations are used in the optimization. For the fluxes, given that we wanted to compare with other approaches, such as the Global Carbon Project estimates, we have run the optimized ORCHIDEE model over a longer period to provide a mean estimate over the 2000 decade. Note finally that restricting the period in figure 6 to three years also helps us to see more clearly the improvements in term of seasonal cycle.

*Second response:*
We added in section 3.1.3 that figure 6 only covers the assimilation period.
For the discussion of the fluxes over the period 2000-2009 we change the introduction to figure 10 in order to justify our choice:
"We ran the optimized model over the full 2000s decade in order to compare with aother estimate of the land surface C residual sink from the Global Carbon Project (GCP, Le Quéré et al, 2015) over the same decade."

p41: 36 regions while in text it is 30.

*First response:*

It was a mistake.
Corrected

*Second response:*
No addition

[revised manuscript text omitted]

---

## Author Response (AR3)

**Response:**

**Editor's comments**

Thanks for addressing the comments and making more substantial changes to the manuscript. Unlike the first 3 reviewers, RC4 was not convinced that your paper makes enough of a contribution to warrant publication in GMD. Having reviewed all the comments and your replies, i think that this paper does indeed make a contribution worth publishing, and hence, I am happy to accept the revised manuscript for final publication in GMD.

In reviewing your manuscript, i noticed a few minor corrections which should be made:

Provide a version number for ORCHIDEE in the abstract please.

You use way too many UN-necessary acronyms in the abstract, for no particular reason. The CCDAS and GCP acronyms are defined, but only used once? Why do you define in acronym in an abstract if you only use it once? The LMDz acronym is used without even being defined?

The Rayner et al 2005 paper has no journal in the reference list?

I believe the Raoult et al paper how now been accepted in GMD, you should used the GMD and not GMDD reference.

Double check all your refs please.

Response:
We thank the editor to accept this paper for publication in GMD and for the additional suggestions to improve the manuscript.

- We have dropped in the abstract the un-necessary acronyms: CCDAS, GPP and GCP.

- We have also defined both in the abstract and in the text the LMDz acronyms as follow: "the general circulation model of the Laboratoire de Météorologie Dynamique (LMDz)"

- We now provide a version number for the ORCHIDEE model in the abstract: "version 1.9.5 used for CMIP5 simulations"

- We have checked the references and we made the following changes: we included the journal name for Rayner et al. (2015); we changed the year for Avitabile et al. to 2016; we provide the GMD reference for Raoult et al. (2016) instead of GMDD; we updated Dufresnes et al. (2013) reference;

Finally, we have slightly changed figure 10 in order to improve its readability. The information content of the figure is kept the same; we have only simplified the bar plot for the NEE as there was no need for separated bars for the total NEE given that the total is also clearly depicted by the stacked bars of the three latitude bands.